# VideoWebArena:
# Evaluating Long Context Multimodal Agents with Video Understanding Web Tasks

**Lawrence Jang**[13], **Yinheng Li**[3], **Dan Zhao**[23]
**Charles Ding**[1], **Justin Lin**[1], **Paul Pu Liang**[2], **Rogerio Bonatti**[3], **Kazuhito Koishida**[3]
[1]Carnegie Mellon University, [2]Massachusetts Institute of Technology,
[3]Microsoft

## Abstract

Videos are often used as a unique source of information in learning how to perform tasks in ways different than what text or static imagery can provide. However, many existing agent benchmarks neglect long-context video understanding, instead focusing on text or static image inputs. To bridge this gap, we introduce VideoWebArena (VideoWA), a benchmark for evaluating the capabilities of long-context multimodal agents for video understanding. VideoWA consists of 2,021 web agent tasks based on manually crafted video tutorials, which total almost four hours of content. For our benchmark, we define a taxonomy of long-context video-based agent tasks with two main areas of focus: skill retention and factual retention. While skill retention tasks evaluate whether an agent can use a given human demonstration to complete a task efficiently, factual retention tasks evaluate whether an agent can retrieve instruction-relevant information from a video to complete a task. We find that the best model achieves a 13.3% success rate on factual retention tasks and 45.8% on factual retention QA pairs—far below human success rates of 73.9% and 79.3%, respectively. On skill retention tasks, long-context models perform worse with tutorials than without, exhibiting a 5% performance decrease in WebArena tasks and a 10.3% decrease in VisualWebArena tasks. Our work highlights performance gaps in the agentic abilities of long-context multimodal models when it comes to video understanding and provides as a testbed for the future development of long-context video agents.[1]

## 1 Introduction

Humans often use videos to complete tasks—following tutorials to retrieving information from within one, or across several, videos. As large foundation models are adapted into AI agents Zheng et al. (2024); Wang et al. (2023); Putta et al. (2024), these multimodal agents may also require similar capabilities to understand and process videos in completing various tasks—especially if they need to retrieve and process said multimodal information (e.g., from the web) to complete tasks that require new knowledge. In these scenarios, the ability of models to maintain long-term memory Zhang et al. (2024b), retrieve and use said information Lewis et al. (2020), and adapt to new information continuously Wu et al. (2024c) is critical for complex multi-step tasks.

While videos contain spatial and temporal dynamics that images or text alone may not convey, integrating video input into multimodal models introduces unique challenges (Tang et al., 2024) such as temporal coherence, context retention, or efficient retrieval over lengthy sequences. However, recent advancements in long-context understanding of large video-capable vision language models, such as LLaVaNeXt (Liu et al., 2023), LongVILA (Xue et al., 2024), and Gemini (Google, 2023) have enabled these agentic models to not only process and understand more information than before, including long video understanding, but also generate and act more freely than before. While many

---

[1]Link to code `https://github.com/ljang0/videowebarena/`
 Link to video `https://www.youtube.com/@webarenawarrior` or
`https://drive.google.com/file/d/17DwmsM7KzBWyz1BN1aq7NHDvgcTIrCgx/view?usp=drive_link`.

existing agentic benchmarks test, they focus primarily on text and image modalities for tasks like interactive QA Mialon et al. (2023), computer tasks Bonatti et al. (2024); Deng et al. (2023), etc.

As a result, from an evaluative perspective, there remains a significant gap in existing benchmarks that can comprehensively evaluate the agentic capabilities of these models across diverse multimodal scenarios, particularly those involving video inputs. The requirement for agents to operate across varying modalities and time frames makes developing and properly evaluating long-context multimodal models essential. Existing benchmarks (Wu et al., 2024b; Mangalam et al., 2023; Li et al., 2024; Pătrăucean et al., 2023) often fall short in testing for long-term memory retention and multimodal integration within an agentic workflow, limiting our understanding of how long context multimodal models can generalize and perform in real-world settings as agents.

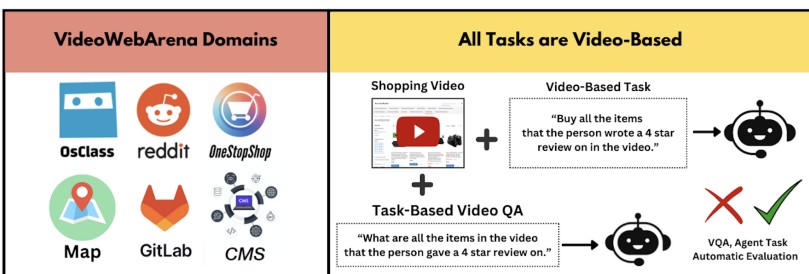

Figure 1: **Overview of VideoWebArena**. VideoWebArena is a visually grounded benchmark that tests the video understanding of agentic models across various realistic domains and environments, mirroring real-life tasks. All tasks require video input and consist of Q/A to test agentic abilities in video information retrieval, video understanding, and more.

**Contributions.** To address this gap, we present a novel, open-source video-based benchmark that evaluates multimodal models' agentic ability to process, understand, and utilize long-context video to accomplish various tasks. In sum, our contributions are:

- We present VideoWebArena—a benchmark focusing specifically on evaluating a model's ability to process long video sequences alongside text and images to complete tasks that require memory retention, information retrieval, multimodal reasoning, and skill retention. VideoWebArena consists of 2,021 tasks and details approximately 4 hours of video content.

- Of these tasks, VideoWebArena consists of 400 factual retention tasks, which test agents' abilities to retrieve information from a video to perform tasks, and 1,621 skill retention tasks, which test agents' abilities to use tutorials in-context to perform tasks more efficiently.

- We evaluate popular and recent video/image-capable state-of-the-art LLMs (e.g., GPT-4o and Gemini 1.5 Pro) to better understand their current long-context video understanding capabilities. Our results show that video/image-capable agents are still limited and far from human levels of performance, highlighting a considerable gap in the information retrieval and agentic abilities of current state-of-the-art long-context models.

The rest of our paper is organized as follows. We begin with a discussion of prior works in Section 2 before detailing the benchmark environment in Section 3. Section 4 describes our experimental setup and Section 5 lays out our benchmarking results. We conclude and discuss our findings in Section 6.

## 2 BACKGROUND

**Large Vision Language Models.** Large vision language models (VLMs) have been popular candidates for incorporating video input understanding. Popular state-of-the-art (SOTA) models like the GPT-4 (OpenAI, 2024) family of models, Claude (AI Anthropic, 2024), and Gemini (Google, 2023) are now able to handle not just text but also visual and even audio input. Similar to LLMs, VLM architectures typically revolve around two types—models with either a joint encoder-decoder architecture such as LLaVA and its variations (Liu et al., 2023) or a decoder-only architecture. Encoder-decoder VLMs tend to project different modalities through a shallow neural network or fully connected layer to link modalities while decoder-style models typically rely on a decoder-only

| Benchmark | Duration (Avg) | Multi-domain? | Multi-skill? | Audio? | Agentic? |
|-----------|----------------|---------------|--------------|--------|----------|
| MSVD-QA (Xu et al. (2017)) | 0.33 | ✓ | ✗ | ✗ | ✗ |
| MSRVTT-QA (Xu et al. (2017)) | 0.25 | ✓ | ✗ | ✗ | ✗ |
| ActivityNet-QA (Yu et al. (2019)) | 1.85 | ✗ | ✗ | ✗ | ✗ |
| NExT-QA (Xiao et al. (2021a)) | 0.73 | ✓ | ✗ | ✗ | ✗ |
| MoVQA (Zhang et al. (2023)) | 16.5 | ✗ | ✓ | ✗ | ✗ |
| MovieChat-1K (Song et al. (2023)) | 9.4 | ✗ | ✓ | ✗ | ✗ |
| MVBench (Li et al. (2023)) | 0.27 | ✓ | ✓ | ✗ | ✗ |
| PerceptionTest (Pătrăucean et al. (2023)) | 0.38 | ✗ | ✓ | ✓ | ✗ |
| CinePile (Rawal et al. (2024)) | 2.7 | ✗ | ✓ | ✓ | ✗ |
| EgoSchema (Mangalam et al. (2024)) | 2.7 | ✗ | ✓ | ✗ | ✗ |
| LongVideoBench (Wu et al. (2024a)) | 7.9 | ✓ | ✓ | ✗ | ✗ |
| Video-MME (Fu et al. (2024a)) | 16.9 | ✓ | ✓ | ✓ | ✗ |
| **VideoWebArena (this work)** | 3.1 | ✓ | ✓ | ✓ | ✓ |

**Table 1:** Comparison of relevant benchmarks/datasets focusing on video modality understanding. Duration denotes the average video duration in minutes, multi-domain denotes if videos cover a diverse range across multiple domains, multi-skill denotes whether the benchmark tests multiple types/aspects of video understanding, audio denotes if the video's audio information is included or used, and agentic denotes whether the benchmark uses video QA as part of an agent's workflow as a tutorial/aid, an input required for completing a multi-step task, etc. We also note that, unlike many benchmarks here, **VideoWebArena** comes with a fully interactive environment, uses open-answer questions relating to video understanding (rather than strictly providing multiple choice options to the model), evaluates task completion on both the final state but also intermediary states, tests across five types/aspects of video understanding grouped into two categories (factual vs. skill retention) and more.

LLM that processes raw inputs (e.g., text tokens, image patches, etc.) such as VILA (Lin et al., 2023) and its variants like VILA[2] (Fang et al., 2024) and X-VILA (Ye et al., 2024). As multimodal understanding with long-context capability becomes more important in processing increasingly large amounts of input information, like video data, models like LongVILA (Xue et al., 2024) can now process a much larger number of video frames in its input for longer videos.

**Agents.** Recent work has turned to adapting large multimodal models like VLMs into autonomous agents, given their capabilities in visual question answering, instruction-following, and high-level reasoning. Many approaches have been developed to improve or supplement the agentic capabilities of these large models, ranging from data collection and synthesis for specific kinds of training (Lai et al., 2024; Patel et al., 2024) or for use at inference time (Pan et al., 2024; Fu et al., 2024b; Koh et al., 2024b; Sarch et al., 2024; Wang et al., 2024). Similar works have shown how certain fine-tuning (Furuta et al., 2024) or prompting (e.g., Set-of-Marks (Yang et al., 2023), (Chi et al., 2024)) can improve performance in navigating web pages to accomplish tasks. Other works have attempted to improve how agents dynamically compose/search for policies themselves (Sodhi et al., 2024).

**Agent Benchmarks.** As more works adapt LLMs and VLMs into agents, there is a need to evaluate their performance. These benchmarks can range across a variety of settings: from general web browsing (Yao et al., 2022; Deng et al., 2023; Zhou et al., 2024; Koh et al., 2024a), where agents are evaluated on their abilities to navigate the web and accomplish specific tasks, to mobile environments, where agents are expected to perform tasks within a mobile OS simulation like Android (Zhang et al., 2024a; Rawles et al., 2024). Other general environments attempt to emulate an OS or computing environment like MMInA (Zhang et al., 2024c), OSWorld (Xie et al., 2024), and Windows Agent Arena (Bonatti et al., 2024) where agents must navigate across multiple computer applications online and offline. Custom environments like WorkArena (Drouin et al., 2024) instead target more specific platforms like ServiceNow in constructing tasks for agent evaluation.

**Long-Context Video Benchmarks.** For large multimodal models, long-context capabilities are essential for detailed planning, action, and understanding, especially for the video modality. There have been multiple benchmarks dedicated towards video understanding, with shorter video inputs (Li et al., 2024), around a few minutes, and longer video inputs (Wu et al., 2024b; Pătrăucean et al., 2023), up to over an hour. Video understanding benchmarks dedicated to temporal reasoning and thematic reasoning (Xiao et al., 2021b; Tapaswi et al., 2016; Lei et al., 2019) also exist. Many of these benchmarks cover subsets and define categories of video understanding tasks related to spatial reasoning, causal reasoning, and temporal reasoning.

| Variable | Value |
|---|---|
| # Videos | 74 |
| Total Duration | 3:48:19 |
| Min Duration | 01:16 |
| Max Duration | 10:41 |
| Avg. Duration | 3:05 |
| Avg. # Factual Retention Tasks per Video | 5.4 |
| Avg. # Skill Retention Tasks per Video | 19.6 |
| Avg. # Videos per Domain | 12.3 |

**Table 2:** Video statistics for VideoWebArena.

| Domain | # Factual Tasks | # Skill Tasks | # Total Tasks |
|---|---|---|---|
| Reddit | 87 (22%) | 206 (13%) | 293 (14%) |
| Classifieds | 60 (15%) | 320 (20%) | 380 (19%) |
| Shopping | 121 (30%) | 654 (40%) | 775 (38%) |
| Shopping (Admin) | 47 (12%) | 182 (11%) | 229 (11%) |
| GitLab | 70 (18%) | 191 (12%) | 261 (13%) |
| Map | 15 (4%) | 68 (4%) | 83 (4%) |
| Total | 400 (100%) | 1621 (100%) | 2021 (100%) |

**Table 3:** Distribution of tasks for VideoWebArena broken down between tasks that test skill retention and factual retention.

# 3 VIDEOWEBARENA ENVIRONMENT

## 3.1 SUMMARY & OVERVIEW

VideoWA centers around six key thematic environments created by VisualWebArena (Koh et al., 2024a) and WebArena (Zhou et al., 2024): Reddit, Classifieds, Shopping, Shopping Admin, Map, and Gitlab. See Tables 2 and 3 for a finer characterization of the tasks and videos within the benchmark.

These domains' websites are locally hosted since the docker images for each website are publicly available online. There is an Amazon Machine Image and instructions dedicated to hosting these websites on an EC2 instance; we refer readers to our codebase for further information. By doing this, we can make our benchmark realistic and reproducible, leveraging data and code from real and popular websites on the internet. We refer readers to WebArena (Zhou et al., 2024) and VisualWebArena (Koh et al., 2024a) for more information on each site and its setup.

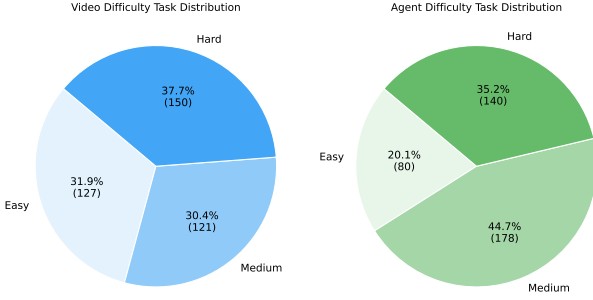

**Figure 2: Left:** VideoWebArena Video Difficulty Task Distribution. **Right:** VideoWebArena Agent Difficulty Task Distribution.

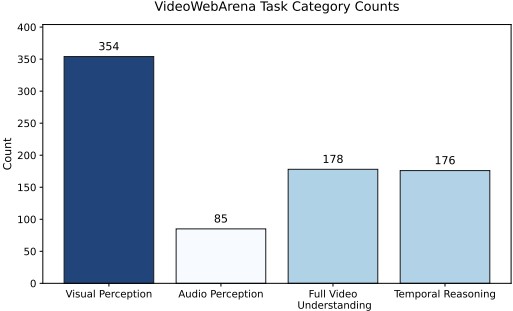

**Figure 3: VideoWebArena Factual Retention Task Counts for Each Category**. The categories are non-exclusive. One task can fall under multiple video perception categories.

| Action Type $a$ | Description |
|---|---|
| click [elem] | Click on element elem. |
| hover [elem] | Hover on element elem. |
| type [elem] [text] | Type text on element elem. |
| press [key_comb] | Press a key combination. |
| new_tab | Open a new tab. |
| tab_focus [index] | Focus on the i-th tab. |
| tab_close | Close current tab. |
| goto [url] | Open url. |
| go_back | Click the back button. |
| go_forward | Click the forward button. |
| scroll [up\|down] | Scroll up or down the page. |
| clear [elem] | Clear a text box element. |
| upload [file path] [elem] | Upload a local file using a upload button. |
| stop [answer] | End the task with an output. |

**Table 4:** List of Action Types and Descriptions

## 3.2 ENVIRONMENT DETAILS

### 3.2.1 FRAMEWORK

We can define an agent's trajectory on our tasks as a partially observable Markov decision process (POMDP) $(S, \mathcal{O}, \mathcal{A}, T, \mathcal{R})$ with state space $S$, observation space $\mathcal{O}$, action space $\mathcal{A}$ containing actions $a$, transition function $T : S \times \mathcal{A} \rightarrow S$, and reward function $\mathcal{R} : S \times \mathcal{A} \rightarrow \mathbb{R}$. Given current observation $o_t \in \mathcal{O}$, an agent generates executable action $a_t \in \mathcal{A}$, resulting in a new state $s_{t+1} \in S$ and a new partial observation $o_{t+1} \in \mathcal{O}$. The reward function $\mathcal{R} : S \times \mathcal{A} \rightarrow [0, 1]$ returns a non-zero value at the final step if the agent state achieves the task objective and zero otherwise. We list available rewards and evaluators in Table 9.

## 3.3 OBSERVATION SPACE

The observation space for our VideoWA environment is predicated on the Set-of-Marks observation space in VisualWebArena. The environment uses executable JavaScript code at each step to extract the interactable HTML elements from the webpage and present them in a top-down order. Similarly, the JavaScript code extracts the bounding boxes of each interactive element and a screenshot of the webpage with bounding boxes over said elements is generated for input into the agent along with the text state representation. At each step, the agent is presented with the overlaid Set-of-Marks screenshot and text observation space, along with the chosen video information to be put into context.

### 3.3.1 ACTION SPACE

We describe the action space of agents within the VideoWA environment in Table 4. Agents are prompted to generate a single action from the action space at each time step. Each action is associated with Playwright Python code that automatically performs the action within the browser. The '[elem]' represents the unique Set-of-Marks element that can be interacted with from the observation space provided through the environment's JavaScript code.

## 3.4 TASK DESIGN

The taxonomy covers two overall task types—skill retention and factual retention—inspired by real-world use cases. We illustrate the taxonomy breakdown in Figure 6. We define skill retention as the ability to learn from and use a given human demonstration to efficiently complete a task For example, using YouTube tutorials or screen recordings of expert demonstrations to learn how to perform a task is a form of skill retention. On the other hand, factual retention is the ability to retrieve information relevant to a user's specific question/task present in a video that may not be the video's main focus (e.g., an incidental detail). For example, one might want to buy the shoes a particular NBA player is wearing but are only shown within a short duration of a much longer basketball highlights video. To complete the task, the model must extract not only information about the specific player but also their shoes, even if this information is secondary to the main content of the video.

| Domain | Video Tutorial | Task Category | Intent | Intermediate Intent |
|---|---|---|---|---|
| **OneStopShop** | *Buy Cheapest Item* | Skill Retention | Buy the cheapest red blanket from Blankets & Throws. | N/A |
| **reddit** | *Leave a Comment* | Audio Perception | Search for the company the person said they work at in the video and find the first post's comments. | What company did the person in the video say they work for? |
| **GitLab** | *How to Star a Repo* | Full Video Understanding | Can you assign the issue with a critical priority label that showed up earliest in the video to @earlev4? | What was the the name of the issue with a critical priority label that showed up earliest in the video? |
| **Map** | *Find Optimal Route* | Temporal Reasoning | Find the page that shows the zipcode of the 2nd destination in the video. | What was the name of the 2nd destination used in the video? |
| **OsClass** | *See Listing Ratings* | Visual Reasoning | Take me to the first red vehicle listing that appears in the video. | What was the name of the first red vehicle listing that appears in the video? |

**Table 5: Examples of Each Task in the VideoWebArena Taxonomy:** Given a video tutorial, the agent is asked to perform the intent. The intermediate intent tests the multimodal agent's ability to extract the necessary information to perform the task from the video. Skill retention tasks do not have intermediate intents as they do not require recalling specific information that factual retention tasks will require.

We present example tasks in Table 5 and an example of an agent on a stylized task in Figure 7. Each task has an 'intent', which is the objective of the task. For all of the newly created factual retention tasks, there is also an 'intermediate_intent', a video-based question that must be answered correctly to have the information necessary to complete the task. Each task also has an automatic evaluator function for both 'intent' and 'intermediate_intent' that returns a score of 0 or 1 based on the environment and response given by the LLM agent. Each task also has an agentic difficulty, distributed between easy, medium, and hard. The agentic difficulty for each task signifies the complexity of the action sequence needed to complete an intent successfully. For agentic difficulty, we classify a task as easy if it can be completed in 1-3 steps, medium if it can be completed in 4-9 steps, and hard if it can be completed in more than 9 steps. This classification is adopted from VisualWebArena (Koh et al., 2024a). Figure 2 provides a detailed breakdown of task difficulty.

## 3.5 VIDEO CREATION AND SKILL RETENTION TASKS

Our benchmark contains 74 unique videos, totaling almost 4 hours of video content (see Table 2 for details)—all of our video tutorials are based on tasks in WebArena and VisualWebArena. We provide our videos online through a YouTube channel and a Google Drive link containing the zip file of all the videos. We formulated these videos by accumulating all the feasible intent templates in WebArena and VisualWebArena. We take 297 unique templates from VisualWebArena and 220 unique templates from WebArena, totaling 1,621 total intents. Further details can be found in Appendix A.1

We map each of our video tutorials to the respective tasks in the WebArena and VisualWebArena task set to create skill retention tasks. We then create 400 original factual retention tasks based on these same tutorials. We had three of the paper's authors create videos and corresponding tasks for each

video they created. We then conducted cross-validation quality assurance with an author who did not make the video/tasks to ensure the task was understandable and able to be completed. We conduct human performance tests similarly, having an author who didn't create the video or tasks attempt the task and have it evaluated by a third author. Further details on task creation and human evaluation can be found in A.2 and A.3.

## 3.6 FACTUAL RETENTION TASKS

For factual retention, we further divide this category into four finer sub-categories: (1) Visual Perception (OCR, Spatial Reasoning), (2) Audio Perception, (3) Full Video Understanding (i.e., tasks that require information across several parts of the video), and (4) Temporal Reasoning (i.e., tasks that require understanding the video with respect to time). One key difference between the factual and skill retention tasks is the intermediate intent and evaluation we create for each factual retention task. The intermediate intent is the video-based question that must be answered correctly to have the information necessary to complete the task. This is intended to decouple the evaluation of agentic abilities in long context video models for video information retrieval tasks; by checking if the model can extract information necessary to complete the task from the video and evaluating that separately from the agent's success, this process can pinpoint the failure modes of the model, whether they come from generating agent actions or video processing.

Additionally, we provide video difficulty ratings for all intermediate intents: easy, medium, and hard. The difficulty ratings signify the complexity of returning the correct answer for a given task's intermediate intent. Easy tasks require returning one piece of information and can be solved with less than 3 frames, medium tasks require returning/retrieving 2 to 3 pieces of information and can be solved with less than half the video, and hard tasks require returning more than 3 pieces that require watching more than half the video. We provide a breakdown Table 5. VideoWA contains 111 unique intent templates across the 400 intents in the factual retention task set.

## 3.7 TASK EVALUATION

Each task has a final and intermediate 'eval' function. We import the automatic functionality from VisualWebArena (Koh et al., 2024a) and WebArena (Zhou et al., 2024) to evaluate our agent tasks. For the intermediate intent evaluation, we use the string-based existing functionality evaluators to assess the agent's response to the video-based question.

All tasks have a final evaluation function (i.e., evaluator) that determines an agent's reward on each task. The reward is typically binary, returning zero or unity depending on whether the agent performs the task unsuccessfully or successfully. Reward values are determined by evaluating the state of the environment at the end of the agent's trajectory to determine if said state matches the correct state corresponding to the correct task execution.

## 4 BASELINE AGENTS

We evaluate our benchmark using three different types of baseline agents with multimodal models as a backbone. Each type is distinguished by the type of video input provided to the model/agent. At each step, the agent is given the task objective, 2 in-context examples, current state $s$, and the input video to the objective as context to generate one action. We describe each type in detail below.

### 4.1 VIDEO IN-CONTEXT AGENT

We define a video input agent that takes the video in at every time step to generate actions. We provide the whole video in-context to the model with the Gemini model. The Gemini model automatically processes the audio, eliminating the need to process audio separately. The specific prompts we use are in Appendix E. We use Set-of-Marks on the website HTML page, the Set-of-Marks element tree string, and the prompt along with the video as input to the model.

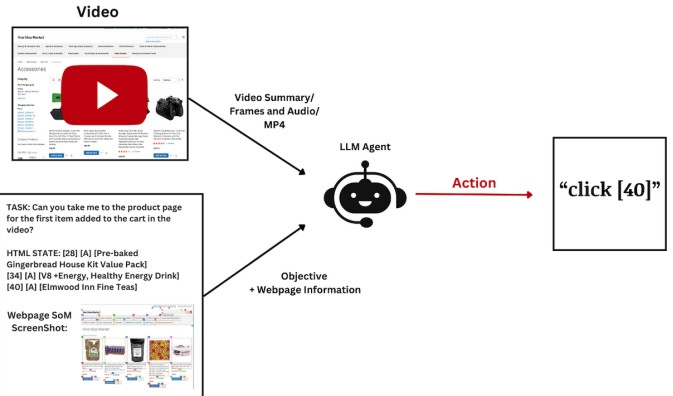

**Figure 4: VideoWebArena Baseline Agent Framework:** We use 3 baseline agents: 1.) Video Summary Agent, where the video summary is fed in-context. 2.) Video Frame Agent, where a set number of frames and audio transcription is fed in-context. 3.) Video Agent, where the video is fed in as an .mov file in-context. The video information is put in-context along with the Set-of-Marks state representation to generate a singular action, following the multimodal SoM agent in VisualWebArena (Koh et al., 2024a).

## 4.2 VIDEO FRAMES IN-CONTEXT AGENT

We define a video input agent that takes a set amount of video frames at every time step along with the video audio to generate actions. To obtain the information from the video, we follow the practice from (Wu et al., 2024b). We sample 1 frame per second (max 60 frames) for the video and include them into the context for the LLM. In addition, we use OpenAI's Whisper (Radford et al., 2022) to transcribe the audio and append it to the context. In this way, we can still pass the information from the video to our LLM. This may not be a perfect method as the video information remains missing during framing sampling. However, since most LLMs on the market only support image and text input, it is essential to experiment with this setting. We use GPT-4o, and the prompt can be seen in Appendix E. We again use Set-of-Marks on the website HTML page, the Set-of-Marks element tree string, and the prompt along with 60 video frames and audio transcriptions as input to the model.

## 4.3 VIDEO SUMMARY IN-CONTEXT AGENT

We define a summary in-context agent that takes a video summary related to the objective at hand in-context at every time step to generate actions. To obtain this summary, we call GPT-4o and feed the video using 60 frames and the Whisper transcription into the model and prompt it to summarize the video concerning the task at hand. Again, our prompt can be seen in Appendix E. Similarly, the summary agent also uses Set-of-Marks for the observation space and generates actions in the aforementioned action space.

## 5 RESULTS

### 5.1 MODEL PERFORMANCE

From Tables 6, 7, and 8, we see varying degrees of agentic performance across the video-capable Gemini and GPT family of models; however, we note several consistent trends across LLM agent results. We outline failure modes in Appendix B. There is no best performing baseline agent or model across skill and factual retention tasks. For factual retention tasks, the summary agent performs the best in task success at 13.3% while the 30 and 100 Frame GPT-4o Agent perform the best in intermediate intent success at 45.8%. For skill retention, we see that long-context models with tutorials actually perform significantly worse than models without tutorials, suggesting that tutorials introduce negative noise that hurt action selection. Although intermediate scores tend to be higher than final scores, this did not translate to task success. This is a constant failure mode of the long-context agents, as they can perform the necessary VQA to extract the necessary information for the task at hand but fall short due to hallucinations, action grounding, and high-level planning errors. For example, in Figure 5, the LLM agent successfully identifies the item to buy from the video. Still, it

| Model | Task Domain | Final Score | Intermediate Score | # Steps (Avg) |
|---|---|---|---|---|
| Gemini 1.5 Pro Video Agent | Classifieds | 6.7% | 41.7% | 17.1 |
| | Gitlab | 5.7% | 35.7% | 18.5 |
| | Map | 6.7% | 73.3% | 9.9 |
| | Reddit | 3.4% | 39.0% | 18.2 |
| | Shopping (admin) | 8.5% | 48.9% | 23.7 |
| | Shopping | 10.0% | 24.7% | 21.6 |
| | Total | 7.0% | 37.0% | 19.4 |
| GPT4-o Summary Agent | Classifieds | 10.0% | 40.0% | 9.7 |
| | Gitlab | 14.2% | 34.7% | 13.0 |
| | Map | 26.7% | 66.7% | 3.8 |
| | Reddit | 11.5% | 39.0% | 13.8 |
| | Shopping (admin) | 8.5% | 29.1% | 13.7 |
| | Shopping | 15.7% | 33.8% | 14.3 |
| | Total | **13.3%** | 36.8% | **12.8** |
| GPT4-o Frame Agent (30 Frames) | Classifieds | 18.3% | 46.6% | 9.3 |
| | Gitlab | 5.7% | 50.0% | 11.8 |
| | Map | 26.7% | 73.3% | 4.7 |
| | Reddit | 6.9% | 42.5% | 11.6 |
| | Shopping (admin) | 8.5% | 57.4% | 16.8 |
| | Shopping | 12.4% | 37.2% | 19.5 |
| | Total | 11.0% | **45.8%** | 14.0 |
| GPT4-o Frame Agent (60 Frames) | Classifieds | 10.0% | 30.0% | 9.5 |
| | Gitlab | 5.7% | 55.7% | 13.4 |
| | Map | 26.7% | 60.0% | 3.5 |
| | Reddit | 2.3% | 44.8% | 11.2 |
| | Shopping (admin) | 4.3% | 48.9% | 13.6 |
| | Shopping | 5.0% | 38.0% | 16.9 |
| | Total | 6.0% | 43.5% | 13.0 |
| GPT4-o Frame Agent (100 Frames) | Classifieds | 13.3% | 41.6% | 7.64 |
| | Gitlab | 7.1% | 58.6% | 14.8 |
| | Map | 20.0% | 53.3% | 3.8 |
| | Reddit | 5.7% | 43.7% | 11.6 |
| | Shopping (admin) | 8.5% | 51% | 14.4 |
| | Shopping | 10.7% | 38.8% | 16.4 |
| | Total | 9.5% | **45.8%** | 13.0 |
| Phi-3.5V (30 Frames) | Classifieds | 0.0% | 8.3% | 8.2 |
| | Gitlab | 2.9% | 10.0% | 6.9 |
| | Map | 0.0% | 26.7% | 16.9 |
| | Reddit | 0.0% | 14.1% | 7.5 |
| | Shopping (admin) | 2.1% | 12.8% | 12.2 |
| | Shopping | 0.08% | 5.8% | 8.6 |
| | Total | 1.2% | 13.0% | 10.1 |
| Human Performance | Classifieds | 61.5% | 69.2% | 7.9 |
| | Gitlab | 81.3% | 81.3% | 7.1 |
| | Map | 69.2% | 76.9% | 4.8 |
| | Reddit | 81.8% | 86.4% | 9.0 |
| | Shopping (admin) | 68.4% | 73.7% | 5.1 |
| | Shopping | 75.0% | 82.1% | 5.0 |
| | Total | 73.9% | 79.3% | 6.4 |

**Table 6: Results on VideoWebArena Factual Retention Tasks.** Performance of GPT4-o, Gemini 1.5 Pro, Phi-3.5V, and human performance on 400 factual retention tasks broken down by task domain. Final scores indicate the overall task performance (i.e., if the task is completed successfully in its entirety), while intermediate scores measure the performance on the intermediate intents.

| Model | WebArena Final Score | Steps | VisualWebArena Final Score | Steps |
|---|---|---|---|---|
| GPT4-o (No Tutorial) | 14.9% | - | 19.8% | - |
| GPT4-o Summary Agent (Tutorial) | 13.8% | 13.9 | 11.6% | 12.4 |
| GPT4-o Frame Agent (Tutorial) | 9.9% | 11.4 | 9.5% | 12.5 |
| Human Performance (No Tutorial) | 82.6% | 12.0 | 72.7% | 12.4 |
| Human Performance (Tutorial) | **93.1%** | **6.1** | **88.6%** | **8.2** |

**Table 7: Results on VideoWebArena Skill Retention Tasks.** Overall performance comparison of GPT4-o and human performance on skill retention tasks. Human performance shows tutorials should help task performance success and efficiency. However, adding tutorials in-context to the model does not necessarily help, but in fact hurts performance by a significant margin. See the failure modes in Appendix B for more analysis. Dashes (-) indicate that data is unavailable for that particular metric.

does not successfully plan and complete the intent. We tested on a smaller subset of tasks with the GPT4-o agent and tested on the full set of tasks with the Gemini agent due to compute constraints.

| Task Category | GPT-4o Summary | GPT-4o (30 Frames) | GPT-4o (60 Frames) | GPT-4o (100 Frames) | Gemini 1.5 Pro | Phi-3.5V (30 Frames) |
|---|---|---|---|---|---|---|
| Visual Perception Task Success Rate | **14.1%** | 11.1% | 6.8% | 9.3% | 7.7% | 0.8% |
| Audio Perception Task Success Rate | 14.8% | **18.1%** | 7.7% | 12.5% | 11.1% | 1.1% |
| Full Video Understanding Task Success Rate | **15.5%** | 10.0% | 7.2% | 10.5% | 6.5% | 1.1% |
| Temporal Reasoning Task Success Rate | **13.7%** | 12.4% | 6.2% | 10.4% | 8.8% | 0.5% |
| Agentic Easy Task Success Rate | **19.5%** | 12.8% | 9.0% | 13.0% | 8.3% | 0.0% |
| Agentic Medium Task Success Rate | **14.2%** | 13.4% | 5.7% | 9.4% | 7.7% | 1.6% |
| Agentic Hard Task Success Rate | **10.8%** | 8.1% | 6.2% | 9.1% | 6.9% | 0.7% |
| Visual Perception Intermediate Success Rate | 32.7% | **43.9%** | 43.0% | 43.5% | 34.0% | 7.9% |
| Audio Perception Intermediate Success Rate | 50.0% | 60.2% | 62.8% | 62.5% | **67.9%** | 23.5% |
| Full Video Understanding Intermediate Success Rate | 34.2% | 40.0% | 40.9% | **41.2%** | 26.2% | 7.3% |
| Temporal Reasoning Intermediate Success Rate | 35.9% | 50.5% | **50.9%** | 50.0% | 38.9% | 8.0% |
| Video Easy Intermediate Success Rate | 39.5% | 52.9% | 52.2% | **53.2%** | 47.1% | 17.3% |
| Video Medium Intermediate Success Rate | 39.4% | 46.2% | **50.4%** | 48.3% | 46.6% | 9.0% |
| Video Hard Intermediate Success Rate | 32.2% | **42.4%** | 40.7% | 41.0% | 26.1% | 5.3% |

Table 8: **Factual Retention Results Breakdown:** Performance breakdown of baseline agents across all task categories and difficulties in the factual retention set. The summary agent has the best task performance, even without having any visual aspect of the video in context. However, it lags behind in the intermediate VQA intents, as the video frame and video agents all perform very similarly on intermediate tasks.

## 5.2 HUMAN PERFORMANCE

To understand the level of human performance expected on the tasks within VideoWA, three authors attempted the tasks and provide intermediate answers for a random sample of each unique task template in the factual retention set. Details on the human evaluation set can be found in Appendix A.3. They also tested 74 unique skill retention tasks, with each task having 2 separate humans attempt the task: one with a tutorial and one without. They performed actions and recorded steps using the VideoWA action space, achieving a success rate of 73.9% on factual retention tasks while only taking an average of 6.4 steps per task (Table 6). Additionally, the intermediate intent and intent performance are linearly correlated while LLMs are not, citing a deficiency in the agentic abilities of these models. For skill retention tasks, human performance registers 93.1% on WebArena tasks and 88.6% in VisualWebArena with tutorials, and 82.6% and 72.7% without tutorials. Naturally, we see a drop in human performance and efficiency without tutorials. In terms of both task success rates and average number of steps taken, video-capable LLM agents lag behind significantly, further emphasizing the need to improve agentic reasoning with video capacity in today's models.

## 6 DISCUSSION

We present VideoWebArena, a video-based agent benchmark that tests the video capability of long-context multimodal models through an agentic lens, consisting of 2,021 tasks that are all video-based, along with 74 manually created videos. These tasks follow a custom defined taxonomy of video-based tasks, providing a wide coverage of task types including skill retention and factual retention to create a comprehensive test bed.

From our experiments, we see that introducing video can result in "noise" on skill-retention tasks, causing the agent to fail at following action generating instructions. For instance, the agent would return try to click using the name of the element rather than the ID number. Another issue was where the agent would generate multi-action responses when the prompt explicitly says to generate one action. From analyzing agent trajectories, we see that agents show inconsistencies in both their reasoning and planning, such as getting stuck in a loop and repeating the same action/step even when the observed state or information does not change. We leave details to Appendix B. We hypothesize that further improvements can result from better distillation of multimodal chain of thought reasoning in the presence of an increasing number of modalities or having the agent learn which modalities are the most relevant for certain tasks over others.

Overall, baseline agents do not perform well on most tasks, falling far behind human performance. It is clear that there is still a long ways ahead in developing intelligent agents capable of effectively utilizing and understanding video in completing regular tasks. As such, further analysis of failure modes from different video agent architectures on our benchmark is key. We hope our environment/benchmark can help facilitate improvement and additional work on improving long-context multimodal agents.

REPRODUCIBILITY STATEMENT

The authors are committed to making this work reproducible. Our code is will be open-sourced and available on Github. An anonymized version of our code base can be found at https://anonymous.4open.science/r/videowebarena-236E/README.md. Our videos will also be made available through Google Drive and YouTube. Our data details are provided in Section 3 and our models and prompts are specified in Section 4.

ETHICS STATEMENT

Our benchmark is intended for safe and responsible innovations of video-based LLM agents. With the rising popularity of LLM agents and the excitement around their deployment, measures to ensure their safe practical deployment and use cases must be present. The authors are committed to the ethical development of LLM agents. For our paper, we did not use human subjects, find any potentially harmful insights, or any ethical concerns. Our benchmark is a self-contained environment to test agents on synthetic tasks.

We would also like to be cognizant of and make aware the possible biases that are introduced in the video tutorials that we have created. Our main focus is studying the ability of agents to make use of video information to accomplish tasks in dynamic settings. As the issues of multi-modal video-capable agent performance and in-context video understanding pose huge obstacles, we made this the main focus of our video and benchmark creation. As all the videos are in English and created by graduate students in an American university, our videos may unintentionally contain possible linguistic and cultural biases. To help mitigate this, we plan on adding Korean/Chinese/Portugese versions of the video dataset as well. Given the significant time and labor required to implement new websites and the main focus of our paper, adding non-English websites unfortunately is beyond the scope of this paper.

We have, however, made it so that our benchmark and its videos can be easily accessed and spun-up by the open-source community to test/study potential biases. We have intentionally left room for updates and extensions to the dataset to incorporate additional underrepresented content or address gaps identified by the community. Our work and its documentation make it very easy for users and the open-source community to create their own tasks customized with their own videos, websites, etc. that reflect specific variations of interest that may be underrepresented here. Existing websites can be used to create new tasks. We plan on monitoring our benchmark's release and usage in the open-source community to address or mitigate possible biases as they arise or are made aware to us through changes in our benchmark or otherwise.

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

## A VIDEOWEBARENA DATA DETAILS

### A.1 VIDEO CREATION DETAILS

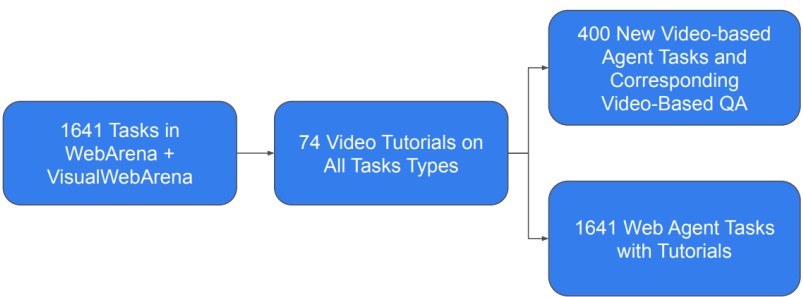

**Figure 5: Dataset Creation Process** A walkthrough of the VideoWebArena dataset creation. From 1641 existing tasks in WebArena and VisualWebArena, the authors grouped these tasks by their intent templates. For each intent template, the authors created a new video tutorial showing how to perform the tasks. For each video, the authors made at minimum 4 factual retention tasks. This led to 1641 skill retention and 400 factual retention tasks.

Three authors of the paper created 74 original video tutorials with audio narrating the actions taken. The three authors evenly divided the videos based off the site the video was based on. We based these 74 tutorials off of 1641 intents in WebArena and VisualWebArena. Each of these tasks are mapped to a video, creating a skill retention task. Each video creator manually checked all of the tasks that they were to create a tutorial for before, then made sure the functionality of the task was shown in the tutorial. We post these videos on YouTube at `https://www.youtube.com/@webarenawarrior`. We also provide them online at a Google Drive link: `https://drive.google.com/file/d/17DwmsM7KzBWyz1BN1aq7NHDvgcTIrCgx/view?usp=drive_link`.

### A.2 TASK CREATION DETAILS

Every video is mapped to minimum one VisualWebArena or WebArena task, creating skill retention tasks. The authors of each video were also tasked with creating a minimum of four factual retention tasks per video, with one task type each from the factual retention taxonomy. The taxonomy can be seen in Figure 6. The authors of each task also are tasked with creating intermediate questions for the factual retention tasks that test if the model can extract the information necessary to complete the task. The authors also create evaluation functions for the intermediate intent and task. Once created, a second author verified and completed each task for quality assurance purposes.

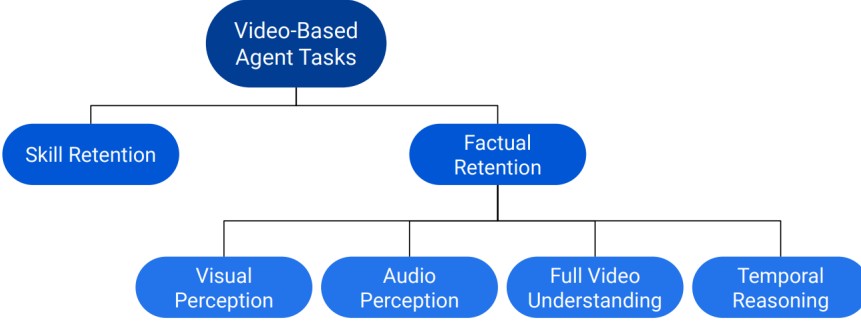

**Figure 6: VideoWebArena Task Taxonomy** We define a taxonomy for all the tasks in our benchmark, namely splitting them into a factual and skill retention groups. Under the factual retention group, there are 4 types of tasks: Visual Perception, Audio Perception, Full Video Understanding, and Temporal Reasoning.

| Evaluator Functions | Reward Condition |
|---|---|
| exact_match($a, \hat{a}$) | 1 if $a$ is exactly $\hat{a}$. |
| must_include($a, \hat{a}$) | 1 if $a$ is in the set $\hat{a}$. |
| fuzzy_match($a, \hat{a}$) | 1 if $a$ and $\hat{a}$ are deemed semantically equal by an LLM. |
| must_exclude($a, \hat{a}$) | 1 if $a$ is not in the set $\hat{a}$. |
| eval_vqa(img, question, $\hat{a}$) | 1 if the output of VQA_Model(img, question) contains $\hat{a}$. |
| eval_fuzzy_image_match(img, $\hat{\text{img}}$) | 1 if the SSIM (Wang et al., 2004) between img, $\hat{\text{img}}$ is higher than a given threshold. |

**Table 9: List of VideoWebArena evaluator functions and descriptions:** All rewards are binary. We adopt our evaluators from WebArena (Zhou et al., 2024) and VisualWebArena (Koh et al., 2024a).

### A.3 HUMAN EVALUATION DETAILS

We conducted two sets of human evaluation, one for the skill retention and one for the factual retention tasks. For the factual retention tasks, an author who did not make the task was given the video, along with the intermediate intent and task intent. Each author was given the agent action space and recorded their number of actions as defined by the agent action space. The answers to the intermediate and task intent were then verified by another author. We did human evaluation on a subset of the factual retention tasks, simply taking a random sample of each unique intent template. We had 111 factual retention tasks for human evaluation. For the skill retention tasks, two authors who did not make the original tutorial were tasked with completing a skill retention task. One author was given the video tutorial before and the other author was not. They then recorded their action steps and completed the task, which was then evaluated and verified by a third author. We did skill retention human evaluation on a singular task per tutorial, totaling to 74 human evaluation skill retention tasks. Given the extremely high success rate for both types of tasks, many of the failures came from human carelessness or interpretation mistakes.

### A.4 ENVIRONMENT DETAILS

We provide a table of the VideoWebArena reward functions in Table 9. These are adapted from VisualWebArena (Koh et al., 2024a).

## B FAILURE MODES

### B.1 COMMON LLM AGENT FAILURE MODES

Many of the basic failures captured in the baseline agents were common repeats of agent errors seen in other agent academic benchmarks. These include hallucinations, where the agent produces a nonsensical action unrelated to its context or task at hand. We attribute this to the lack of instruction tuning and model alignment on agentic tasks. Another common failure mode displayed in the baseline agents was failure to do visual grounding. The agents will recognize the correct plan of action, but choose the wrong element with respect to the Set-of-Marks image input and take the wrong action.

Action grounding and planning was also a common failure mode of the baseline agents. An agent can simply generate the wrong plan or action that will yield unsuccessful trajectories, and not change this plan even with negative feedback from the environment. This suggests using inference time search or memory based methods can be effective to combat these failures. Incorporating self-reflection during inference can also help the agents recover from failures in action grounding and planning. The lack of self-reflection is especially seen when the agent generates the same action repetitively, leading the task to terminate. Even though an action is shown to be unsuccessful towards completing a task, an agent will continue to repeatedly attempt the same action to try and complete the task.

### B.2 LONG-CONTEXT SPECIFIC FAILURE MODES

Within our skill and factual retention tasks, there were many failure modes that presented issues relevant to long-context modeling. One constant issue we noticed was failure to adhere to the prompt instructions for generating actions. With the extra noise provided with the video information in-context, the agent did not always adhere to the action generation guidelines provided in the prompt.

For example, under the Set-of-Marks elements, a click action must be generated using `click [elem]` where elem is the numeric ID of the SoM element. However, the agent would return `click [elem]` where elem was the name of the element. This formatting issue persisted for other actions with the longer prompt.

A common issue for skill retention tasks was the agent began generating multi-action responses when the prompt explicitly says to generate one action. Given the tutorial or summary to complete a similar task, the agent would get distracted by the comprehensive plan and generate multiple actions from the video information, straying away from the prompt guidelines. This led to failure to complete tasks.

A common issue for factual retention tasks was video grounding. Specifically, we could pinpoint that the video-frame and summary agents would simply miss visual information due to the nature of their video processing. Additionally, the video agent also showcased many of these video grounding errors. For example, a common task was to `Take me to the page in the video when event` happened. However, if the frames or summary did not include this page, there was no way for the agent to get to this page or know about its existence. This issue was exacerbated in tasks that required full video understanding or temporal reasoning across the video. This is a flaw in the baseline agent setup we proposed. Many of the audio tasks were completed at a much higher rate than the video perception tasks, citing that video grounding is a larger issue than audio grounding when processing these modalities within videos. We encourage better video understanding agent systems with our benchmark.

## C    TASK CREATION DOCUMENTATION

### C.1    EXAMPLE TASK JSON

We provide an example task below in their JSON file format.

```
{
    "sites": [
        "classifieds"
    ],
    "task_id": 4,
    "require_login": true,
    "storage_state": "./.auth/classifieds_state.json",
    "start_url": "__CLASSIFIEDS__",
    "geolocation": null,
    "intent_template": "Can you take me to {description}?",
    "intent": "Can you take me to the cheapest primary red boat
        for sale in the video?",
    "intermediate_intent": "What was the name of the post for the
        cheapest primary red boat for sale in the video?",
    "video": "search_sort_comment",
    "instantiation_dict": {
        "description": "the cheapest primary red boat for sale in
            the video"
    },
    "require_reset": false,
    "intermediate_eval": {
        "eval_types": [
            "string_match"
        ],
        "reference_answers": {
            "must_include": [
                "1986 nova xl 23ft"
            ]
        },
        "reference_url": "",
        "program_html": [],
```

```
        "string_note": "",
        "reference_answer_raw_annotation": ""
    },
    "eval": {
        "eval_types": [
            "url_match"
        ],
        "reference_answers": null,
        "reference_url": "__CLASSIFIEDS__/index.php?page=item&id=4
            9894",
        "program_html": [],
        "url_note": "EXACT"
    },
    "intermediate_difficulty": "hard",
    "overall_difficulty": "medium",
    "visual_perception": true,
    "audio_perception": false,
    "full_video_understanding": true,
    "temporal_reasoning": false,
    "comments": ""
}
```

## C.2  TASK CREATION DOCUMENTATION

We used the following instructions to task creators. These can be used to create new tasks. Additionally, you can use any video as input to your LLM agent - our evaluation suite simply chooses from our own 74 manually made tutorials.

In order to create a task, choose or make a video of your own to reference to in order to create a task. Each task should be video-grounded, such that the task can only be completed after watching the video. For each new task, you need to fill the following values in a json task file:

1. **Sites:** The site(s) used in the task.
2. **Start URL:** The url the task starts on.
3. **Video:** The video name associated with the task.
4. **Task ID:** A unique ID for the task.
5. **Intent:** The prompt for the video-based task.
6. **Intent Template:** The task template specific to the intent with default parameters.
7. **Intermediate Eval:** Evaluates if your Vision-Language Model (VLM) can extract the necessary information from the video through a task-based VQA. More documentation on existing evaluators can be found in the evaluators.py file in our repository.
8. **Eval:** The evaluation suite to test whether the agentic task was completed successfully. More documentation on existing evaluators can be found in the evaluators.py file in our repository.
9. **Intermediate Difficulty:** The difficulty level of the intermediate VQA intent, distributed between easy, medium, and hard. The video difficulty ratings signify the complexity of returning the correct answer for a given task's intermediate intent. Easy tasks require returning one piece of information and can be solved with less than 3 frames, medium tasks require returning 2 to 3 things and can be solved with less than half the video, and hard tasks require returning more than 3 things and require watching more than half the video.
10. **Overall Difficulty:** The overall difficulty level of the task, distributed between easy, medium, and hard. The agentic difficulty for each task signifies the complexity of the action sequence needed to complete an intent successfully. For agentic difficulty, we classify a task as easy if it can be completed in 1-3 steps, medium if it can be completed in 4-9 steps, and hard if it can be completed in more than 9 steps

11. **Visual Perception:** Whether visual perception of the video input is required. True or False.

12. **Audio Perception:** Whether audio reasoning of the video input is required. True or False.

13. **Full Video Understanding:** Whether full video comprehension of the video input is necessary. True or False.

14. **Temporal Reasoning:** Whether reasoning across different time points in the video input is required. True or False.

# D RESULTS

## D.1 ADDITIONAL RESULTS

We provide another result breakdown plot at Table 10. This shows the average steps per task type.

**Table 10:** Model Comparison - Average Steps per Task Type

| Category | GPT-4o Summary | GPT-4o (30 Frames) | GPT-4o (60 Frames) | GPT-4o (100 Frames) | Gemini 1.5 Pro |
|---|---|---|---|---|---|
| Visual Perception | **12.9** | 14.5 | 13.7 | 13.2 | 19.7 |
| Audio Perception | 10.5 | 10.5 | **10.2** | 10.5 | 17.0 |
| Full Video Understanding | **12.5** | 14.4 | 13.6 | 13.0 | 20.3 |
| Temporal Reasoning | 14.9 | 14.6 | **13.9** | 14.5 | 20.1 |
| Agentic Easy | 12.6 | **10.1** | **10.1** | 10.2 | 20.6 |
| Agentic Medium | **11.6** | 14.7 | 13.5 | 12.5 | 19.2 |
| Agentic Hard | 14.3 | 15.6 | **14.9** | 15.4 | 19.4 |
| Video Easy | **11.8** | 13.4 | 11.9 | 13.4 | 19.5 |
| Video Medium | 13.1 | 13.2 | 12.9 | **11.8** | 19.5 |
| Video Hard | **13.2** | 15.5 | 14.6 | 13.8 | 19.6 |

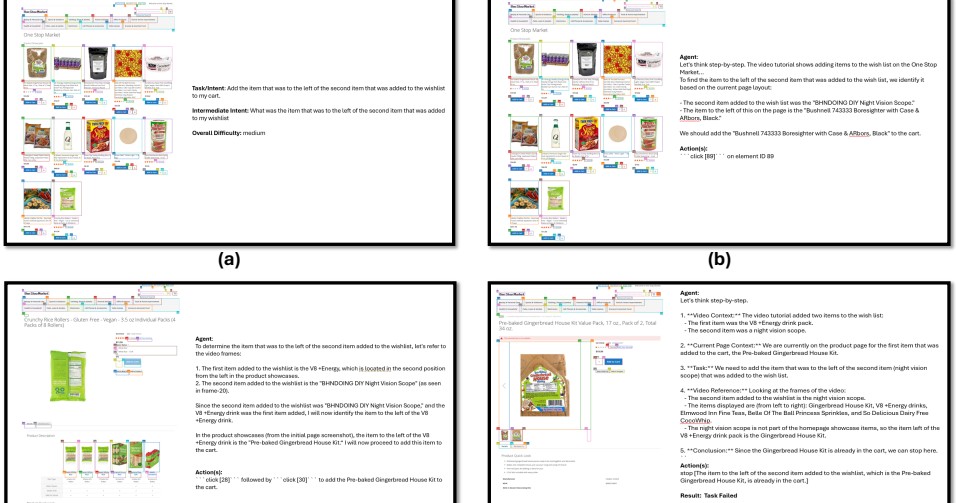

**Figure 7: Abridged Example of VideoWebArena Task.** A stylized example of a task in VideoWebArena: starting from (a) to (b), the task is defined, and an agent interacts with its visual input to create a plan and perform actions. From (b) to (c), it continues its actions and planning along its trajectory for the task before concluding (incorrectly) in (d), where it receives a final reward of zero for failing to complete the task correctly.

# E AGENT PROMPTS

## E.1 VIDEO AGENT TASK PROMPT

You are an autonomous intelligent agent tasked with navigating a web browser. You will be given web-based tasks that can be done based on information in a video. These tasks will be accomplished through the use of specific actions you can issue.

Here's the information you'll have:
1. The user's objective: This is the task you're trying to complete.
2. A video tutorial about this task or a similar task will be provided to assist you.
3. The current web page's accessibility tree: This is a simplified representation of the webpage, providing key information.
4. The current web page's URL: This is the page you're currently navigating.
5. The open tabs: These are the tabs you have open.
6. The previous action: This is the action you just performed. It may be helpful to track your progress.

The actions you can perform fall into several categories:
# Page Operation Actions
```click [id]```: This action clicks on an element with a specific id on the webpage.
```type [id] [content]```: Use this to type the content into the field with id. By default, the "Enter" key is pressed after typing unless press_enter_after is set to 0, i.e., ```type [id] [content] [0]```.
```hover [id]```: Hover over an element with id.
```press [key_comb]```: Simulates the pressing of a key combination on the keyboard (e.g., Ctrl+v).
```scroll [down]``` or ```scroll [up]```: Scroll the page up or down.
# Tab Management Actions
```new_tab```: Open a new, empty browser tab.
```tab_focus [tab_index]```: Switch the browser's focus to a specific tab using its index.
```close_tab```: Close the currently active tab.
# URL Navigation Actions
```goto [url]```: Navigate to a specific URL.
```go_back```: Navigate to the previously viewed page.
```go_forward```: Navigate to the next page (if a previous 'go_back' action was performed).
# Completion Action
```stop [answer]```: Issue this action when you believe the task is complete. If the objective is to find a text-based answer, provide the answer in the bracket.

Homepage:
If you want to visit other websites, check out the homepage at http://homepage.com. It has a list of websites you can visit.
http://homepage.com/password.html lists all the account name and password for the websites. You can use them to log in to the websites.

To be successful, it is very important to follow the following rules:
1. You should only issue an action that is valid given the current observation.
2. You should only issue one action at a time.
3. You should follow the examples to reason step by step and then issue the next action.
4. Generate the action in the correct format. Start with a "In summary, the next action I will perform is" phrase, followed by action inside ```. For example, "In summary, the next action I will perform is ```click [1234]```."
5. Issue stop action when you think you have achieved the objective. Don't generate anything after stop.

Example 1:

VIDEO: Shopping Tutorial MOV File
OBSERVATION: [28] [A] [Pre-baked Gingerbread House Kit Value Pack, 17 oz., Pack of 2, Total 34 oz.] [] [StaticText] [19.99] [30] [BUTTON] [Add to Cart] [34] [A] [V8 +Energy, Healthy Energy Drink, Steady Energy from Black and Green Tea, Pomegranate Blueberry, 8 Ounce Can, Pack of 24] [] [StaticText] [14.47] [36] [BUTTON] [Add to Cart] [40] [A] [Elmwood Inn Fine Teas, Orange Vanilla Caffeine-free Fruit Infusion, 16-Ounce Pouch] [] [StaticText] [19.36] [42] [BUTTON] [Add to Cart] [43] [A] [Add to Wish List]
URL: http://onestopmarket.com
OBJECTIVE: Can you take me to the product page for the first item added to the cart in the video?
PREVIOUS ACTION: None
OUTPUT: Let's think step-by-step. This page is the home page for the One Stop Market and includes the first item added to the cart in the video, which was Elmwood Inn Fine Teas, Orange Vanilla Caffeine-free Fruit Infusion, 16-Ounce Pouch. In the video tutorial, it shows that clicking on the title of the listing will take me to the item page. Therefore, I will issue the click action with the item listing. In summary, the next action I will perform is ```click [40]```.

Example 2:

VIDEO: Reddit Tutorial MOV File
OBSERVATION: [StaticText] [Forums] [1] [A] [Forums] [9] [A] [Alphabetical] [] [StaticText] [allentown] [10] [A] [allentown] [] [StaticText] [baltimore] [16] [A] [baltimore] [] [StaticText] [books] [17] [A] [books] [] [StaticText] [boston] [18] [A] [boston] [] [StaticText] [MachineLearning] [52] [A] [MachineLearning] [] [StaticText] [pittsburgh] [78] [A] [pittsburgh] [] [StaticText] [technology] [90] [A] [technology] [] [StaticText] [television] [91] [A] [television] [] [StaticText] [Running Postmill] [105] [A] [Postmill]
URL: http://reddit.com
OBJECTIVE: Can you take me to the post in the video the user commented on?
PREVIOUS ACTION: None
OUTPUT: Let's think step-by-step. This page lists all the forums on the website. In the video, the post that the user left a comment was under the r/technology forum, so I should navigate to that. I can navigate to that forum by first clicking on the r/technology link. Therefore, I will issue the click action. In summary, the next action I will perform is ```click [90]```.

## E.2 VIDEO FRAME AGENT TASK PROMPT

You are an autonomous intelligent agent tasked with navigating a web browser. You will be given web-based tasks that can be done based on information in a video. These tasks will be accomplished through the use of specific actions you can issue.

Here's the information you'll have:
1. The user's objective: This is the task you're trying to complete.
2. A video tutorial about this task or a similar task will be provided to assist you.
3. The current web page's accessibility tree: This is a simplified representation of the webpage, providing key information.
4. The current web page's URL: This is the page you're currently navigating.
5. The open tabs: These are the tabs you have open.
6. The previous action: This is the action you just performed. It may be helpful to track your progress.

The actions you can perform fall into several categories:
# Page Operation Actions
```click [id]```: This action clicks on an element with a specific id on the webpage.
```type [id] [content]```: Use this to type the content into the field with id. By default, the "Enter" key is pressed after typing unless press_enter_after is set to 0, i.e., ```type [id] [content] [0]```.
```hover [id]```: Hover over an element with id.
```press [key_comb]```: Simulates the pressing of a key combination on the keyboard (e.g., Ctrl+v).
```scroll [down]``` or ```scroll [up]```: Scroll the page up or down.
# Tab Management Actions
```new_tab```: Open a new, empty browser tab.
```tab_focus [tab_index]```: Switch the browser's focus to a specific tab using its index.
```close_tab```: Close the currently active tab.
# URL Navigation Actions
```goto [url]```: Navigate to a specific URL.
```go_back```: Navigate to the previously viewed page.
```go_forward```: Navigate to the next page (if a previous 'go_back' action was performed).
# Completion Action
```stop [answer]```: Issue this action when you believe the task is complete. If the objective is to find a text-based answer, provide the answer in the bracket.

Homepage:
If you want to visit other websites, check out the homepage at http://homepage.com. It has a list of websites you can visit.
http://homepage.com/password.html lists all the account name and password for the websites. You can use them to log in to the websites.

To be successful, it is very important to follow the following rules:
1. You should only issue an action that is valid given the current observation.
2. You should only issue one action at a time.
3. You should follow the examples to reason step by step and then issue the next action.
4. Generate the action in the correct format. Start with a "In summary, the next action I will perform is" phrase, followed by action inside ```. For example, "In summary, the next action I will perform is ```click [1234]```."
5. Issue stop action when you think you have achieved the objective. Don't generate anything after stop.

Example 1:

VIDEO FRAMES: 5 Frames from a Shopping Tutorial
AUDIO: Hi everyone, welcome to a tutorial on the One Stop Market. Today this is just a general tutorial video and how to get around on things. So one thing you need to get to an item is simply click on the title or the image. And as you can see here is going to take me to the title. From here I can edit the quantity, add the cart, add to my wish list, add to my comparisons, and I can access through views here. Similarly if I go to this I will be very similar, this one has 12 views so I can see 12 views, I can also leave my own review at the bottom here. And then if I want to add items to my cart and just click add to cart, if I want to add to my wish list I can click the red heart button. Similarly I add a cart, sometimes you are going to get prompted with an option to add items details before I add the cart. Other times it is not going to be an option like here, add to your comparison page here as well. And so if I want to go to different sections I can go here, let's go to Xbox One, let's go and have a look. There are also subsections within categories, so these are also categories, so accessories is also a category. Let's find the most expensive item. And you can do this by sorting my price and then flipping the arrow, it says to my cart, let's get it in black, and then similarly you can go to my cart here. I'm going to go to view it on a cart and we can see that our cart is here and if I want to go back to the One Stop Market this is how things go. So I hope this helps and thanks for watching.
OBSERVATION: [28] [A] [Pre-baked Gingerbread House Kit Value Pack, 17 oz., Pack of 2, Total 34 oz.] [] [StaticText] [19.99] [30] [BUTTON] [Add to Cart] [34] [A] [V8 +Energy, Healthy Energy Drink, Steady Energy from Black and Green Tea, Pomegranate Blueberry, 8 Ounce Can, Pack of 24] [] [StaticText] [14.47] [36] [BUTTON] [Add to Cart] [40] [A] [Elmwood Inn Fine Teas, Orange Vanilla Caffeine-free Fruit Infusion, 16-Ounce Pouch] [] [StaticText] [19.36] [42] [BUTTON] [Add to Cart] [43] [A] [Add to Wish List]
URL: http://onestopmarket.com
OBJECTIVE: Can you take me to the product page for the first item added to the cart in the video?
PREVIOUS ACTION: None
OUTPUT: Let's think step-by-step. This page is the home page for the One Stop Market and includes the first item added to the cart in the video, which was Elmwood Inn Fine Teas, Orange Vanilla Caffeine-free Fruit Infusion, 16-Ounce Pouch. In the video tutorial, it shows that clicking on the title of the listing will take me to the item page. Therefore, I will issue the click action with the item listing. In summary, the next action I will perform is ```click [40]```.

Example 2:

VIDEO FRAMES: 5 Frames from a Reddit Tutorial
AUDIO: I wanted to make a quick tutorial on how to use the reddit site. So let's say I wanted to make a response to a comment under one of the top posts under the r/technology forum. So I can click under this forums link here. Scroll down to technology. And let's say I wanted to view its comments so I can click here. And then look at all those comments. And I can see that this is the top comment here. And let's say I wanted to reply great comment. So I can get a quick preview. I can post it. And then now it shows me that I have successfully made a comment under this single comment. Right.

OBSERVATION: [StaticText] [Forums] [1] [A] [Forums] [9] [A] [Alphabetical] [] [StaticText] [allentown] [10] [A] [allentown] []
[StaticText] [baltimore] [16] [A] [baltimore] [] [StaticText] [books] [17] [A] [books] [] [StaticText] [boston] [18] [A] [boston] []
[StaticText] [MachineLearning] [52] [A] [MachineLearning] [] [StaticText] [pittsburgh] [78] [A] [pittsburgh] [] [StaticText] [technology]
[90] [A] [technology] [] [StaticText] [television] [91] [A] [television] [] [StaticText] [Running Postmill] [105] [A] [Postmill]]
URL: http://reddit.com
OBJECTIVE: Can you take me to the post in the video the user commented on?
PREVIOUS ACTION: None
OUTPUT: Let's think step-by-step. This page lists all the forums on the website. In the video, the post that the user left a comment
was under the r/technology forum, so I should navigate to that. I can navigate to that forum by first clicking on the r/technology link.
Therefore, I will issue the click action. In summary, the next action I will perform is ```click [90]```.

## E.3  VIDEO SUMMARY AGENT TASK PROMPT

You are an autonomous intelligent agent tasked with navigating a web browser. You will be given web-based tasks that can be done
based on information in a video. These tasks will be accomplished through the use of specific actions you can issue.

Here's the information you'll have:
1. The user's objective: This is the task you're trying to complete.
2. A summary from a tutorial for a similar task: This provides useful information for solving this task.
3. The current web page's accessibility tree: This is a simplified representation of the webpage, providing key information.
4. The current web page's URL: This is the page you're currently navigating.
5. The open tabs: These are the tabs you have open.
6. The previous action: This is the action you just performed. It may be helpful to track your progress.

The actions you can perform fall into several categories:
# Page Operation Actions
```click [id]```: This action clicks on an element with a specific id on the webpage.
```type [id] [content]```: Use this to type the content into the field with id. By default, the "Enter" key is pressed after typing unless
press_enter_after is set to 0, i.e., ```type [id] [content] [0]```.
```hover [id]```: Hover over an element with id.
```press [key_comb]```: Simulates the pressing of a key combination on the keyboard (e.g., Ctrl+v).
```scroll [down]``` or ```scroll [up]```: Scroll the page up or down.
# Tab Management Actions
```new_tab```: Open a new, empty browser tab.
```tab_focus [tab_index]```: Switch the browser's focus to a specific tab using its index.
```close_tab```: Close the currently active tab.
# URL Navigation Actions
```goto [url]```: Navigate to a specific URL.
```go_back```: Navigate to the previously viewed page.
```go_forward```: Navigate to the next page (if a previous 'go_back' action was performed).
# Completion Action
```stop [answer]```: Issue this action when you believe the task is complete. If the objective is to find a text-based answer, provide the
answer in the bracket.

Homepage:
If you want to visit other websites, check out the homepage at http://homepage.com. It has a list of websites you can visit.
http://homepage.com/password.html lists all the account name and password for the websites. You can use them to log in to the websites.

To be successful, it is very important to follow the following rules:
1. You should only issue an action that is valid given the current observation.
2. You should only issue one action at a time.
3. You should follow the examples to reason step by step and then issue the next action.
4. Generate the action in the correct format. Start with a "In summary, the next action I will perform is" phrase, followed by action inside
```. For example, "In summary, the next action I will perform is ```click [1234]```."
5. Issue stop action when you think you have achieved the objective. Don't generate anything after stop.

Example 1:
VIDEO SUMMARY: The tutorial explains how to navigate the One Stop Market website and manage items. To view an item, simply
click on its title or image, which takes you to a page where you can adjust the quantity, add it to your cart, wish list, or comparison list,
and leave a review. Some items may prompt you to provide details before adding them to your cart. You can also browse different
sections, like Xbox One or Accessories, and sort items by price. After selecting an item and adding it to your cart, you can view your cart
and return to the main marketplace. The video concludes with a note of thanks.
OBSERVATION: [28] [A] [Pre-baked Gingerbread House Kit Value Pack, 17 oz., Pack of 2, Total 34 oz.] [] [StaticText] [19.99] [30]
[BUTTON] [Add to Cart] [34] [A] [V8 +Energy, Healthy Energy Drink, Steady Energy from Black and Green Tea, Pomegranate
Blueberry, 8 Ounce Can, Pack of 24] [] [StaticText] [14.47] [36] [BUTTON] [Add to Cart] [40] [A] [Elmwood Inn Fine Teas, Orange
Vanilla Caffeine-free Fruit Infusion, 16-Ounce Pouch] [] [StaticText] [19.36] [42] [BUTTON] [Add to Cart] [43] [A] [Add to Wish List]
URL: http://onestopmarket.com
OBJECTIVE: Can you take me to the product page for the first item added to the cart in the video?
PREVIOUS ACTION: None
OUTPUT: Let's think step-by-step. This page is the home page for the One Stop Market and includes the first item added to the cart in
the video, which was Elmwood Inn Fine Teas, Orange Vanilla Caffeine-free Fruit Infusion, 16-Ounce Pouch. In the video tutorial, it
shows that clicking on the title of the listing will take me to the item page. Therefore, I will issue the click action with the item listing. In
summary, the next action I will perform is ```click [40]```.

Example 2:

VIDEO SUMMARY: The tutorial explains how to leave a comment on a Reddit post, start by logging into your account. Navigate to the subreddit of your choice, such as r/technology, either by searching or selecting it from your subscribed subreddits. Once there, select a post you want to comment on, and scroll down to view existing comments. If you wish to comment on the post itself, scroll to the bottom where you'll find an Add a comment box. To reply to a specific comment, click the Reply button under that comment. After typing your comment, you can preview it by clicking the Preview button if you'd like to see how it will look. When you're ready, click Post to submit the comment. Your comment should appear immediately beneath the post or the specific comment you replied to.

OBSERVATION: [StaticText] [Forums] [1] [A] [Forums] [9] [A] [Alphabetical] [] [StaticText] [allentown] [10] [A] [allentown] [] [StaticText] [baltimore] [16] [A] [baltimore] [] [StaticText] [books] [17] [A] [books] [] [StaticText] [boston] [18] [A] [boston] [] [StaticText] [MachineLearning] [52] [A] [MachineLearning] [] [StaticText] [pittsburgh] [78] [A] [pittsburgh] [] [StaticText] [technology] [90] [A] [technology] [] [StaticText] [television] [91] [A] [television] [] [StaticText] [Running Postmill] [105] [A] [Postmill]

URL: http://reddit.com

OBJECTIVE: Can you take me to the post in the video the user commented on?

PREVIOUS ACTION: None

OUTPUT: Let's think step-by-step. This page lists all the forums on the website. In the video, the post that the user left a comment was under the r/technology forum, so I should navigate to that. I can navigate to that forum by first clicking on the r/technology link. Therefore, I will issue the click action. In summary, the next action I will perform is ```click [90]```.

## E.4 VIDEO AGENT INTERMEDIATE TASK PROMPT

You are an autonomous intelligent agent that extracts information from videos. You will be given this video and a question. You need to answer the question based on the video provided.

Example 1:

VIDEO: Shopping Tutorial MOV File
QUESTION: What is the first item that gets added to the cart on the One Stop Market in the video?
ANSWER: Elmwood Inn Fine Teas, Orange Vanilla Caffeine-free Fruit Infusion, 16-Ounce Pouch

Example 2:

VIDEO: Reddit Tutorial MOV File
QUESTION: What is the name of the author of the post that the person in the video commented on?
ANSWER: Sorin61

## E.5 VIDEO FRAME AGENT INTERMEDIATE TASK PROMPT

You are an autonomous intelligent agent that extracts information from videos. You will be given a list of frames sampled from a video and its audio transcription. You need to answer the question based on the video provided.

Example 1:

VIDEO FRAMES: 5 Frames from Shopping Tutorial
AUDIO: Hi everyone, welcome to a tutorial on the One Stop Market. Today this is just a general tutorial video and how to get around on things. So one thing you need to get to an item is simply click on the title or the image. And as you can see here is going to take me to the title. From here I can edit the quantity, add the cart, add to my wish list, add to my comparisons, and I can access through views here. Similarly if I go to this I will be very similar, this one has 12 views so I can see 12 views, I can also leave my own review at the bottom here. And then if I want to add items to my cart and just click add to cart, if I want to add to my wish list I can click the red heart button. Similarly I add a cart, sometimes you are going to get prompted with add to add items details before I add the cart. Other times it is not going to be an option like here, add to your comparison page here as well. And so if I want to go to different sections I can go here, let's go to Xbox One, let's go and have a look. There are also subsections within categories, so these are also categories, so accessories is also a category. Let's find the most expensive item. And you can do this by sorting my price and then flipping the arrow, it says to my cart, let's get it in black, and then similarly you can go to my cart here. I'm going to go to view it on a cart and we can see that our cart is here and if I want to go back to the One Stop Market this is how things go. So I hope this helps and thanks for watching.
QUESTION: What is the first item that gets added to the cart on the One Stop Market in the video?
ANSWER: Elmwood Inn Fine Teas, Orange Vanilla Caffeine-free Fruit Infusion, 16-Ounce Pouch

Example 2:

VIDEO FRAMES: 5 Frames from Reddit Tutorial
AUDIO: I wanted to make a quick tutorial on how to use the reddit site. So let's say I wanted to make a response to a comment under one of the top posts under the r/technology forum. So I can click under this forums link here. Scroll down to technology. And let's say I wanted to view its comments so I can click here. And then look at all those comments. And I can see that this is the top comment here. And let's say I wanted to reply great comment. So I can get a quick preview. I can post it. And then now it shows me that I have successfully made a comment under this single comment. Right.
QUESTION: What is the name of the author of the post that the person in the video commented on?
ANSWER: Sorin61

## E.6 VIDEO SUMMARY AGENT INTERMEDIATE TASK PROMPT

You are an autonomous intelligent agent that extracts information from summaries. You will be given a summary of a video and a question about the video. You need to answer the question based on the summary provided.

Example 1:

VIDEO SUMMARY: Let's think step-by-step. To add an item to your cart on the One-Start Market, first navigate to the website and browse or search for the desired item. You can add an item directly by clicking the blue 'Add to Cart' button next to it, which updates the cart icon to reflect the addition. Alternatively, click on the item listing to access its detailed page, where you can select options like size or color and adjust the quantity before adding it to the cart. For example, you can select the size and add eight flannel shirts for your family.

View and edit your cart by clicking the cart emblem/icon, which provides access to all added items and options to proceed to checkout.
QUESTION: What is the first item that gets added to the cart on the One Stop Market in the video?
ANSWER: Elmwood Inn Fine Teas, Orange Vanilla Caffeine-free Fruit Infusion, 16-Ounce Pouch

Example 2:

VIDEO SUMMARY: Let's think step-by-step. To leave a comment on a Reddit post, start by logging into your account. Navigate to the subreddit of your choice, such as r/technology, either by searching or selecting it from your subscribed subreddits. Once there, select a post you want to comment on, and scroll down to view existing comments. If you wish to comment on the post itself, scroll to the bottom where you'll find an Add a comment box. To reply to a specific comment, click the Reply button under that comment. After typing your comment, you can preview it by clicking the Preview button if you'd like to see how it will look. When you're ready, click Post to submit the comment. Your comment should appear immediately beneath the post or the specific comment you replied to.
QUESTION: What is the name of the author of the post that the person in the video commented on?
ANSWER: Sorin61

## E.7 VIDEO SUMMARIZATION PROMPT

You are an autonomous intelligent agent tasked with learning from a video to accomplish a task. You will be given a video. You will be given a task to complete. You will need to extract useful information to accomplish the task.

Example 1:

VIDEO: Shopping Tutorial MOV File
OBJECTIVE: Add an item to the cart on the One Stop Market.
SUMMARY: Let's think step-by-step. To add an item to your cart on the One Stop Market, first navigate to the website and browse or search for the desired item. You can add an item directly by clicking the blue 'Add to Cart' button next to it, which updates the cart icon to reflect the addition. Alternatively, click on the item listing to access its detailed page, where you can select options like size or color and adjust the quantity before adding it to the cart. For example, you can select the size and add eight flannel shirts for your family. View and edit your cart by clicking the cart emblem/icon, which provides access to all added items and options to proceed to checkout.

Example 2:

VIDEO: Reddit Tutorial MOV File
OBJECTIVE: Leave a comment on a Postmill post.
SUMMARY: Let's think step-by-step. To leave a comment on a Reddit post, start by logging into your account. Navigate to the subreddit of your choice, such as r/technology, either by searching or selecting it from your subscribed subreddits. Once there, select a post you want to comment on, and scroll down to view existing comments. If you wish to comment on the post itself, scroll to the bottom where you'll find an Add a comment box. To reply to a specific comment, click the Reply button under that comment. After typing your comment, you can preview it by clicking the Preview button if you'd like to see how it will look. When you're ready, click Post to submit the comment. Your comment should appear immediately beneath the post or the specific comment you replied to.

## E.8 VIDEO FRAME SUMMARIZATION PROMPT

You are an autonomous intelligent agent tasked with learn froming a video to accomplish a task. You will be given a list of frames sampled from a video and its audio transcription. You will be given a task to complete. You will need to extract useful information to accomplish the task.

Example 1:

VIDEO FRAMES: 5 PNG Frames from Shopping Tutorial
AUDIO: Hi everyone, welcome to a tutorial on the One Stop Market. Today this is just a general tutorial video and how to get around on things. So one thing you need to get to an item is simply click on the title or the image. And as you can see here is going to take me to the title. From here I can edit the quantity, add the cart, add to my wish list, add to my comparisons, and I can access through views here. Similarly if I go to this I will be very similar, this one has 12 views so I can see 12 views, I can also leave my own review at the bottom here. And then if I want to add items to my cart and just click add to cart, if I want to add to my wish list I can click the red heart button. Similarly I add a cart, sometimes you are going to get prompted with an option to add items details before I add the cart. Other times it is not going to be an option like here, add to your comparison page here as well. And so if I want to go to different sections I can go here, let's go to Xbox One, let's go and have a look. There are also subsections within categories, so these are also categories, so accessories is also a category. Let's find the most expensive item. And you can do this by sorting my price and then flipping the arrow, it says to my cart, let's get it in black, and then similarly you can go to my cart here. I'm going to go to view it on a cart and we can see that our cart is here and if I want to go back to the One Stop Market this is how things go. So I hope this helps and thanks for watching.
OBJECTIVE: Add an item to the cart on the One Stop Market.
SUMMARY: Let's think step-by-step. To add an item to your cart on the One Stop Market, first navigate to the website and browse or search for the desired item. You can add an item directly by clicking the blue 'Add to Cart' button next to it, which updates the cart icon to reflect the addition. Alternatively, click on the item listing to access its detailed page, where you can select options like size or color and adjust the quantity before adding it to the cart. For example, you can select the size and add eight flannel shirts for your family. View and edit your cart by clicking the cart emblem/icon, which provides access to all added items and options to proceed to checkout.

Example 2:

VIDEO FRAMES: 5 PNG Frames from Reddit Tutorial
AUDIO: I wanted to make a quick tutorial on how to use the reddit site. So let's say I wanted to make a response to a comment under one of the top posts under the r/technology forum. So I can click under this forums link here. Scroll down to technology. And let's say I wanted to view its comments so I can click here. And then look at all those comments. And I can see that this is the top comment here. And let's say I wanted to reply great comment. So I can get a quick preview. I can post it. And then now it shows me that I have successfully made a comment under this single comment. Right. OBJECTIVE: Leave a comment on a Postmill post.
SUMMARY: Let's think step-by-step. To leave a comment on a Reddit post, start by logging into your account. Navigate to the subreddit of your choice, such as r/technology, either by searching or selecting it from your subscribed subreddits. Once there, select a post you want to comment on, and scroll down to view existing comments. If you wish to comment on the post itself, scroll to the bottom where

you'll find an Add a comment box. To reply to a specific comment, click the Reply button under that comment. After typing your comment, you can preview it by clicking the Preview button if you'd like to see how it will look. When you're ready, click Post to submit the comment. Your comment should appear immediately beneath the post or the specific comment you replied to.

