# OpenReview forum: "VideoWebArena:  Evaluating Long Context Multimodal Agents with Video Understanding Web Tasks"
_ICLR.cc/2025/Conference — ICLR 2025 Poster_

### Official Review · Reviewer_pezb · 2024-11-02

**Soundness:** 3
**Presentation:** 3
**Contribution:** 2
**Rating:** 6
**Confidence:** 4

**Summary:**

The authors present VideoWA, a benchmark designed to assess the capabilities of long-context multimodal agents, particularly their ability to understand and utilize video information to accomplish tasks on web-based platforms. The paper emphasizes the gap in existing benchmarks, which predominantly focus on static image or text inputs, and seeks to bridge this with comprehensive video-based task evaluation.

**Strengths:**

1. The paper is clearly written, with well-structured sections that guide the reader through the motivation, methodology, and results.
2. The experimental setup and evaluation in the paper are robust, employing a range of state-of-the-art models such as GPT-4o and Gemini 1.5 Pro. The results are well-documented and provide clear evidence of the limitations of current models in handling long-context multimodal inputs.
3. The significance of this work lies in its potential to drive advancements in the development of long-context multimodal models. Although some tasks may not fully justify the necessity of video tutorials, identifying more suitable tasks that would effectively leverage the richness of video input remains a challenge. Nevertheless, the benchmark's foundation provides a valuable starting point for refining task design and exploring the benefits of video-based learning in AI agents.

**Weaknesses:**

Major problems I have found are as follows:
1. One significant concern is that the contributions of VideoWA appear to be relatively incremental compared to prior work such as WebArena and VisualWebArena. While the inclusion of video content does enhance task complexity and data richness, the paper does not convincingly demonstrate how these video tutorials meaningfully improve agent learning or task performance. Specifically, some tasks, like “buying the cheapest item” as presented in Table 4, do not seem to justify the necessity of video input. Agents could reasonably complete these tasks with textual intent or static images. This raises doubts about whether the chosen tasks fully exploit the potential advantages of video tutorials or merely add unnecessary complexity.
2. Another perplexing issue is the low success rate of human performance on seemingly straightforward tasks. The authors report that these metrics were gathered using the authors themselves, who should be well-acquainted with both the data and the tasks. The review questions how the human success rate could be so limited (e.g., 73.9% on factual retention), suggesting a potential shortcoming in task design or evaluation methodology. If even familiar humans struggle, it may reflect an underlying flaw in the task construction or evaluation criteria, which could compromise the benchmark's real-world applicability and reliability.
3. The paper presents an unexpected and concerning trend where multimodal models, even when provided with video tutorials, perform worse than models without them. This result contradicts the assumption that video tutorials should aid skill retention, as evidenced by the drop in performance on WebArena and VisualWebArena tasks (5% and 10.3% decreases, respectively). The authors should address whether the noise introduced by video information or ineffective agentic reasoning mechanisms is the primary cause of these shortcomings. Without further investigation or hypothesis testing, the paper leaves this critical aspect unresolved.

**Questions:**

Please refer to the weakness 1-3.

---

> ### Author Response · Authors · 2024-11-21
>
> We thank the reviewer for the insightful comments. We address concerns below.
>
> **Concern re: contributions of VideoWA**
>
> * We believe that video as an input modality has value on its own especially given how relatively few evaluative works, as of the time of writing, have been done on how large foundation model-based agents perform with videos, together with text etc., as inputs under dynamic agentic settings:
>   * how an agent model understands an instruction via video vs. text can be considerably different as each modality has different kinds/amounts of informational context.
>   * e.g., humans sometimes prefer to use videos to learn how to do certain tasks (vs reading docs) as they may more "easier" or contain additional information.
> * Our work demonstrates that these video tutorials may not be used/processed optimally by long-context models in the manner humans  use videos to complete tasks, and there are innovations to be made on that front.
> * Re: the necessity of video input; agent performance on the original WebArena and VisualWebArena tasks we provide tutorials for is 14.9% and 19.78%, respectively. Therefore, the agents are not able to perform these tasks at a prolific rate, especially given that human performance sit at 82.6% and 72.7%.
>   * This performance gap inspired the creation of video tutorials and their ability to help the agents. We also show in our human evaluations that providing a tutorial to humans gives a boost over these numbers in success rate and efficiency.
>   * However, our experiments of providing tutorials to agents show they actually perform worse, which is an interesting and possibly new finding.
>
> **Low human success rates**
> * In human evaluations, participants had no prior knowledge of the tasks or any assistance (e.g., internet access, search tools, replay). We consider 73.9% a reasonable success rate and note that many errors came from inconsistent human attentiveness (especially for longer videos, etc.).
> * Considering many of the tasks require watching a full video to complete a task, this can become tedious for human evaluators, which can take over 5 minutes in one contiguous sitting. Several cases also appeared to result from human error (e.g., carelessness).
> * We also note that these success rates are, on average, in line with human performance levels among other agentic benchmarks such as WebArena (~78%), OSWorld (~72%), Windows Agent Arena (~74%), etc.
> * Additionally, given the range of difficulty levels in our benchmark’s tasks, we expected that the non-uniformity of task difficulty will result in less than perfect, or even very high levels of, human performance.
>
>
> **"Noise" introduced by video information or ineffective agentic reasoning**
> * We observed that the "noise" from introducing video causes issues with skill-retention tasks; for example, this, at times, causes the agent to fail at following action generating instructions.
>   * e.g., under the Set-of-Marks elements, a click action must be generated using click [elem] where elem is the numeric ID of the SoM element. However, the agent would return click [elem] where elem was the name of the element. This formatting issue persisted for other actions with the longer prompt.
>   * e.g., a common issue for skill retention tasks was the agent generating multi-action responses when the prompt explicitly says to generate one action.
>   * e.g., Given the tutorial or summary to complete a similar task, the agent would get distracted by the comprehensive plan and generate multiple actions from the video information, straying away from the prompt guidelines. Many times, the agent became distracted by the video and began generating actions unrelated to the task or its previous actions.
> * One hypotheses relates to the backbone of these agents (multi-modal LLMs) as they were first grounded in language and have recently been adapted to other modalities, resulting in "imperfect" or lossy learning of, and across, them.
>   * As such, instead of being first grounded in vision  (e.g., learning/pre-training with images), this adaptation to image/video via its language/text understanding may introduce “noise” in its learning, causing unwarranted and unexpected behavior.
> * From analyzing agent trajectories, we found that agents show inconsistencies in both their reasoning and planning.
>   * e.g., agents can get stuck in a loop, repeating the same action/step even when the observed state or information does not change. Agents also undo correct tasks or partial progress or simply hallucinate.
>   * Additionally, similar issues with visual grounding/understanding of the webpage remains an issue seen even in VisualWebArena; the agent will generate a correct plan but choose the wrong element to interact with.
> * We will include a section with more details of failure modes in the revised manuscript.
>
> Please let us know of any other questions!

---

> > ### Author Response · Authors · 2024-11-24
> > **Follow-Up to Rebuttal**
> >
> > Dear reviewer, thank you again for your valuable feedback. Since the rebuttal period is ending soon, we were wondering whether our changes have addressed your concerns. Please let us know and we will be happy to engage further.

---

> > > ### Comment · Reviewer_pezb · 2024-11-26
> > >
> > > Dear authors, thank you for your rebuttal. Yes your rebuttal has addressed most of my concerns.

---

### Official Review · Reviewer_QoA2 · 2024-11-03

**Soundness:** 3
**Presentation:** 2
**Contribution:** 3
**Rating:** 6
**Confidence:** 3

**Summary:**

This paper presents the VideoWebArena benchmark to evaluate the capabilities of long context multimodal agents. The tasks are divided into factual retention and skill retention. The authors evaluate GPT-4o and Gemini-1.5-pro on their benchmarks and provide an in-depth analysis.

**Strengths:**

+ The proposed benchmark is valuable to the community.
+ The task designs make sense to me (divide into skill retention and factual retention), which challenge current modes’ capabilities to extract information from a demonstration video to complete tasks.
+ The experimental findings are insightful, especially Table 7, which suggests that current models struggle with “learning” from tutorial videos, in contrast to humans.

**Weaknesses:**

+ The introduction lacks references.

+ While the related works section reviews a few agent benchmarks and long-context benchmarks, it is suggested to include a dataset comparison table of VideoWA with existing benchmarks. This would clarify VideoWA’s distinctiveness, highlighting differences in task diversity, domain, video length, etc., against existing ones.

+ To improve clarity, it would be helpful for the authors to explicitly discuss how VideoWebArena differs from VisualWebArena and WebArena in terms of environment and task design. Are there methodological innovations in this paper beyond extending these benchmarks to long videos?

+ Could the authors further clarify the difference among video agent, frames agent and summary agent in terms of their input? Does the difference lie in audio input (gemini directly takes audio while gpt-4o takes audio transcriptions)? What about the summary agent?

+ In Table 4, the example provided for temporal reasoning appears more like a counting task; substituting it with a clearer example focused on temporal reasoning might improve clarity.

+ The benchmark only evaluates Gemini and GPT-4o. I wonder why the authors do not include open-source VLMs such as LongVILA, and LLAVANextVideo (as mentioned in the intro).

**Questions:**

Overall, this paper offers a useful benchmark with valuable insights for the community.  However, the current presentation is not clear and many clarifications are needed. Please refer to weaknesses for my questions.

---

> ### Author Response · Authors · 2024-11-21
>
> We thank the reviewer for their insightful comments and suggestions. We address concerns below.
>
> **References in introduction**
> * We will add references to better contextualize our paper, focusing on the development of video-capable and multi-modal agents as well as their benchmarks.
>
> **Dataset comparison table of existing benchmarks**
> * Great suggestion. We will include a table highlighting a comparison of related benchmarks in our revised manuscript in Section 2 so readers can better visualize and compare benchmarks with ours.
>
> **Differences b/w VideoWebArena and VisualWebArena and WebArena**
> * First, although we use similar environments as VisualWebArena/WebArena do, we introduce *intermediate evaluation* into our task design for our bechmark.
>   * As noted in our Task Design section, each video-based task has an intermediate question/answer pair that must be answered correctly for the task to be completed successfully.
>   * e.g., if the video-based task is “Block the user that made the post about reindeers in the video”, the intermediate question would be “What was the name of the user that made the post about reindeers in the video”. **All of our 400 factual retention tasks include this intermediate intent, that was manually created and verified by the papers’ authors.**
> * Second, we create **~4 hours of video content for tutorials/tasks;** while we use these for our benchmark, this considerable amount of content can also be used for various modeling, planning, and behavior cloning applications in WebArena/VisualWebArena, larger web agent studies, and more.
>
> **Differences among video/frames/summary agent in terms of their input**
> * In short, the video agent takes in a MP4 or MOV file (without extracting audio), while the video frame agent takes in *N* frames and the audio transcription of the video using Whisper (OpenAI). We include these details in the Baseline Agents section of the paper.
> * In comparison, the summary agent takes in a summary of the video with respect to the task as well as the summary text at every step.
>   * The summary is generated by feeding the video or the video frames + audio as input and prompting the VLM to give a summary of the task at hand. Sample prompts are in the Appendix.
>
> **In Table 4: a clearer example focused on temporal reasoning**
> * Thank you for pointing this out. We will instead change the example to a task in the Gitlab domain, "Can you assign the issue with the critical priority label which showed up earliest in the video to @earlev4?". This task does not explicitly require counting and hopefully highlights the temporal nature of the task!
>
> **Inclusion of open-source VLMs such as LongVILA, and LLAVANextVideo**
> * Due to memory/compute resource limits, as well as the short timeframe for discussion period (with experiments for even a closed-source model taking ~2weeks), we unfortunately have been unable to test large open-source models. Examples include:
>     * Memory: Many commercial APIs that host open-source multi-modal models support multi-modal input but have restrictions on input format. For example, on Azure, both LLaMA3 vision and Phi3 vision only accept a single image input (videos need several images/frames); hoever, we require ~60 images per inference.
>       * Others like Replicate limit input size to 20MB per request while our inputs can easily exceed 50MB due to the video content.
>     * Compute: Our experiments are also computationally intensive and time-consuming as they involve processing up to 60 images, along with other content, per inference.
> * As such, we test a smaller open-source multi-modal model, Phi-3.5V using 30 frames. Its predecessor Phi3 has also shown strong performance in long video understanding, at times better than 7B/13B models (source: Long Video Bench https://arxiv.org/pdf/2407.15754).
>   * Results of Phi-3.5V are below which will be included in our revised manuscript as they serve as a good contrast to our existing results (Phi-3.5V is a smaller open-sourced multi-modal model vs. Gemini and GPT which are large closed-source ones).
>
> | Domain          	| Score (success rate)  | Score (intermediate) | Avg. # Steps 	| # Tasks  | Total Score  | Total Intermediate Score |
> |---------------------|---------|--------------------|-----------|------------|--------------|---------------------------|
> | classified 	| 0   	| 0.083         	| 8.217 	| 60     	| 0        	| 5                     	|
> | gitlab     	| 0.029   | 0.100         	| 6.857 	| 70     	| 2        	| 7                     	|
> | map        	| 0   	| 0.267         	| 16.933	| 15     	| 0        	| 4                     	|
> | reddit     	| 0   	| 0.141         	| 7.482 	| 87     	| 0        	| 12                    	|
> | shopping   	| 0.008   | 0.058         	| 8.608 	| 121    	| 1        	| 7                     	|
> | shopping (admin) | 0.021   | 0.128         	| 12.255	| 47     	| 1        	| 6                     	|
>
>
> Please let us know if any other questions/concerns arise.

---

> > ### Author Response · Authors · 2024-11-24
> > **Follow-Up to Rebuttal**
> >
> > Dear reviewer, thank you again for your valuable feedback. Since the rebuttal period is ending soon, we were wondering whether our changes have addressed your concerns. Please let us know and we will be happy to engage further.

---

> > > ### Comment · Reviewer_QoA2 · 2024-11-26
> > >
> > > Thank the authors for providing the responses and additional experiments with Phi3. I appreciate the effort, however, I feel that some of my key points have not been adequately clarified.
> > >
> > > I would like to point out that during this discussion phase, the authors have the option to upload a revised version of the paper. A revised manuscript could help address many of my concerns. While the responses mention “we will add” certain clarifications, no concrete updates to the paper have been provided.
> > >
> > > In addition, regarding the differences with prior works such as VideoWebArena and VisualWebArena, the response does not sufficiently address the technical novelty of the proposed approach. This leaves the concern of incremental contributions unresolved, as similarly shared by reviewer pezb.

---

> > > > ### Author Response · Authors · 2024-11-27
> > > >
> > > > Thank you for your response.
> > > >
> > > > Per the reviewers’ question/concern on the inclusion of open-source models -- we have run experiments with phi3.5-V which, on our benchmark, **represents a significant request and is a significantly resource/time intensive set of experiments especially given the the relatively short length of the discussion window**.
> > > >
> > > > We apologize for the wait — however, we have focused our efforts on completing/running these requested experiments and are currently verifying these results for errors etc. We had planned on submitting the revised manuscript once all our changes were finalized.
> > > >
> > > > ## Manuscript Revisions
> > > > We have just finished and uploaded a revised version of our manuscript with some of our stated changes, including but not limited to:
> > > >
> > > > * Quality of life changes throughout the paper (e.g., fixing typos, adding content and revisions to improve flow/clarity)
> > > > * Phi-3.5V Model results (Section 5, Tables 6 and 8),
> > > > * Dataset Comparison table (Section 2, Table 1),
> > > > * Revised Table 5 with a more relevant Temporal Reasoning Example
> > > > * Revised and improved introduction (with more context and references)
> > > > * Revised and improved discussion section
> > > > * Revised ethics statement to address possible biases and mitigation efforts
> > > > * Added Appendix B on more details of failure modes, task documentation creation, etc.
> > > >
> > > > We noticed that you lowered your score after our first response -- we would be happy to discuss what caused this despite the additional experiments, revisions, etc. performed to improve the paper on behalf of the reviewer’s request. Please let us know if there are concerns, suggestions, or questions and we’d be happy to address them.
> > > >
> > > > ## Technical/Methodological Novelty
> > > > Regarding technical/methodological novelty, we want to emphasize that:
> > > > * We manually create a considerable amount of original video content and test large foundation models’ video understanding and utilization through an agentic lens which, to our knowledge, has not been done before.
> > > > * We rely not on just final state evaluation (e.g., WebArena/VisualWebarena) but also on intermediate intent/evaluation; this means we decouple the evaluation of how well agents do at task information retrieval and generating agentic actions.
> > > > * Unlike many other video benchmarks, we do not use multiple choice questions, but rather allow open-form answering by agents in order to test/understand their ability to in a more complex agentic space.
> > > > * Our results via our ablations, show how different ways of parsing or formatting informational content with videos affect agent performance along with several performance gaps for video-capable agents that go beyond just video QA.
> > > > * Please also see our response to **Reviewer pezb’s** concerns as well as **Reviewer pezb’s** recent reply to us.
> > > >
> > > > Thank you again and please don’t hesitate to let us know if anything else!

---

> > > > > ### Comment · Reviewer_QoA2 · 2024-12-03
> > > > >
> > > > > I appreciate the authors’ efforts in updating the manuscript. After reviewing the updates and other comments, I have adjusted my score back to 6.

---

> ### Author Response · Authors · 2024-12-03
>
> Thank you again for taking the time to review our changes/revisions -- we very much appreciate the reviewer's time and feedback in helping improve our work!

---

### Official Review · Reviewer_a1Hy · 2024-11-03

**Soundness:** 3
**Presentation:** 3
**Contribution:** 3
**Rating:** 8
**Confidence:** 3

**Summary:**

This paper present VideoWebArena (benchmark) to evaluate a model's ability to process long video sequences alongside text and images to complete tasks that require memory retention, information retrieval, multimodal reasoning, and skill retention. Moreover, this papaer show that these models are still a far reach from human levels of performance, highlighting a wide gap in the information retrieval and agentic abilities of current state-of-the-art long-context models.

Strengths:
+ This paper construct a benchmark called VideoWebArena.
+ The idea of this paper is novel and interesting.
+ This paper test many retention tasks, I think it is a hard task and the author complete this task.

Weakness:
+ Can author release the source code for this paper, I would like to try this agent.

**Strengths:**

See Summary

**Weaknesses:**

See Summary

**Questions:**

See Summary

---

> ### Author Response · Authors · 2024-11-21
>
> We are very encouraged that the reviewer finds our benchmark and agent experiments novel and interesting!
>
> **Source code for this paper**
> * Yes, we plan on making our benchmark and agent code available with robust documentation and details regarding installation, setup, and implementation along with examples and common problems/solutions. However, due to ICLR’s double-blind policy, our current repo is private. We will make our repo public immediately afterwards.
>
> Please let us know of any other concerns/questions!

---

> > ### Comment · Reviewer_a1Hy · 2024-11-22
> > **To authors**
> >
> > I look forward to the author releasing the code after the manuscript is finalized. I think this is an excellent paper and deserves to be included in ICLR. Thanks again!

---

### Official Review · Reviewer_4THA · 2024-11-04

**Soundness:** 3
**Presentation:** 3
**Contribution:** 3
**Rating:** 6
**Confidence:** 3

**Summary:**

This paper presents VideoWebArena, a novel, open-sourced video-based benchmark, designed to evaluate the capabilities of long-context multimodal agents in video understanding tasks. The dataset consists of 2,021 tasks based on four hours of video tutorials across six domains. Moreover, the paper conducts experiments to validate that the current intelligent agents do not perform well on most the task, and is important to the relevant research field.

**Strengths:**

1.	This paper provides a novel video benchmark that is very welcome to the research community.
2.	The benchmark offers a well-defined taxonomy that focuses on two main areas: factual retention and skill retention, which test different facets of a model's abilities to retrieve information from videos and efficiently apply learned skills.
3.	The paper conducts experiments to validate that the current intelligent agents do not perform well on most the task, leading to future improvement.

**Weaknesses:**

1.	The used LLMs are all closed-source, thus may cause obstacles to reproduction. Experiments on open-sourced intelligent agents may be included.
2.	This paper does not provide a comparison or discussing over itself and other similar video benchmarks, like MVBench[1], with respect to some tasks like temporal reasoning, etc.

[1] Li, K., Wang, Y., He, Y., Li, Y., Wang, Y., Liu, Y., ... & Qiao, Y. (2024). Mvbench: A comprehensive multi-modal video understanding benchmark. In Proceedings of the IEEE/CVF Conference on Computer Vision and Pattern Recognition (pp. 22195-22206).

**Questions:**

1.	Can new tasks or new domain be easily added into the benchmark?
2.	Through analysis, what is the possible aspects that the existing agent can be improved?

---

> ### Author Response · Authors · 2024-11-21
>
> We thank the reviewer for their insightful questions. We address your concerns below.
>
> **Adding testing/evaluation of open-sourced models**
>
> * Given our focus on constructing, scaling, and testing the benchmark’s tasks and video creation, our efforts on model evaluation prioritized popular state-of-the-art foundation models that take in video modality as input (e.g., GPT, Gemini) as well as their ablations.
> * Due to limits on memory and compute, as well as the short timeframe for discussion period (with experiments for even a closed-source model taking ~2weeks), we unfortunately have been unable to test large open-source models. Examples include:
>    * Memory: Many commercial APIs that host open-source multi-modal models support multi-modal input but have restrictions on input format. For example, on Azure, both LLaMA3 vision and Phi3 vision only accept a single image input (videos need several images/frames); hoever, we require ~60 images per inference. Others like Replicate limit input size to 20MB per request while our inputs can easily exceed 50MB due to the video content.
>   * Compute: Our experiments are also computationally intensive and time-consuming as they involve processing up to 60 images, along with other content, per inference.
> * As such, we test a smaller open-source multi-modal model, Phi-3.5V. Its predecessor Phi3 has also shown strong performance in long video understanding, at times better than 7B/13B models (source: Long Video Bench https://arxiv.org/pdf/2407.15754).
>
> * Results of Phi-3.5V are below which will be included in our revised manuscript as they serve as a good contrast to our existing results (Phi-3.5V is a smaller open-sourced multi-modal model vs. Gemini and GPT which are large closed-source ones).
>
> | Domain          	| Score (success rate)  | Score (intermediate) | Avg. # Steps 	| # Tasks  | Total Score  | Total Intermediate Score |
> |---------------------|---------|--------------------|-----------|------------|--------------|---------------------------|
> | classified 	| 0   	| 0.083         	| 8.217 	| 60     	| 0        	| 5                     	|
> | gitlab     	| 0.029   | 0.100         	| 6.857 	| 70     	| 2        	| 7                     	|
> | map        	| 0   	| 0.267         	| 16.933	| 15     	| 0        	| 4                     	|
> | reddit     	| 0   	| 0.141         	| 7.482 	| 87     	| 0        	| 12                    	|
> | shopping   	| 0.008   | 0.058         	| 8.608 	| 121    	| 1        	| 7                     	|
> | shopping (admin) | 0.021   | 0.128         	| 12.255	| 47     	| 1        	| 6                     	|
>
> **Lack of discussion of other video benchmarks**
> * We will include a fuller discussion and comparison of other benchmarks, including MVBench, to highlight differences and contextualize the contributions of each.
> * Our main difference/novelty/contribution lies in the agentic nature of our video benchmark, which requires an agent to generate actions in an interactive environment on an open-ended task in addition to answering video-understanding-based questions.
>
>
> **Adding new tasks/domains to the benchmark**
> * Yes, new tasks/domains can be easily added. We will include instructions on creating new tasks, along with examples, in our readme and repo which will give a comprehensive walkthrough.
> * Adding new domains is also very easy simply by specifying the starting URL in the task JSON. We will also add an example task json to the appendix to further clarify.
>
> **Possible aspects of improving existing agents**
> * Our analysis of failure modes found similar agent errors also seen in other benchmarks:
>   * Hallucinations, failures in visual grounding, infinite loops, undoing correctly done tasks, and failing to adhere to the prompt instructions for generating actions.
>     * e.g., under the Set-of-Marks elements, a click action must be generated using click [elem] where elem is the numeric ID of the element. However, the agent would return the name of the element.
>     * e.g., the agent generated multi-action responses when the prompt explicitly instructed the agent to generate one action. When given the tutorial/summary to complete a similar task, the agent became distracted by the summary or overall task plan and began generating multiple actions from the video information.
>
> * Regarding improvement, we propose several possibilities given our observations:
>   * Better distill/coordinate multimodal chain of thought reasoning so that models can improve reasoning and planning performance in the presence of an increasing number of modalities.
>   * Teach the agent to prioritize relevant/mandatory instructions regarding the task over less important, dense in-context information.
>   * Have the agent learn when to invoke the video modality (or any modalities): if some modalities are helpful for certain tasks, the agent should then extract/process that information.
>
> Please let us know of any other questions!

---

> > ### Author Response · Authors · 2024-11-24
> > **Follow-Up to Rebuttal**
> >
> > Dear reviewer, thank you again for your valuable feedback. Since the rebuttal period is ending soon, we were wondering whether our changes have addressed your concerns. Please let us know and we will be happy to engage further.

---

> ### Comment · Area_Chair_2YTG · 2024-11-27
>
> Dear reviewer,
>
> Today is the last day for reviewers to ask questions to authors. Did the authors' rebuttal address your concern? Do you have any additional questions?

---

> > ### Comment · Reviewer_4THA · 2024-11-27
> > **Response to the rebuttal**
> >
> > Thank the authors for the rebuttal. I will maintain my score.

---

### Official Review · Reviewer_pi6N · 2024-11-18

**Soundness:** 3
**Presentation:** 3
**Contribution:** 3
**Rating:** 5
**Confidence:** 2

**Summary:**

This paper introduces VideoWebArena, a benchmark designed to evaluate multimodal AI models’ abilities to process and understand long video sequences alongside text and images for completing tasks. As AI assistants increasingly need to understand video inputs to perform workflows, learn skills, and make decisions autonomously, challenges arise with maintaining temporal coherence, long-term memory, and information retrieval over extended sequences. VideoWebArena addresses a gap in existing benchmarks by focusing on these long-context multimodal capabilities, comprising 2,021 tasks with approximately 4 hours of video content. The benchmark includes 400 factual retention tasks that test information retrieval and 1,621 skill retention tasks that assess the use of in-context tutorials. Evaluations of advanced video-capable models, like GPT-4o and Gemini 1.5 Pro, show that while they exhibit basic capabilities with video content, significant gaps remain compared to human-level understanding, particularly in long-term memory and task execution. This benchmark provides a critical tool for advancing and assessing long-context video comprehension in multimodal AI.

**Strengths:**

It introduces VideoWebArena, a comprehensive benchmark designed specifically to evaluate the long-context understanding and multimodal reasoning of models, addressing a gap in existing evaluation tools. The benchmark includes a wide range of tasks (2,021 in total), split between factual retention and skill retention tasks, which test both memory retrieval and the application of learned information.

The study provides valuable insights by evaluating prominent video-capable LLMs like GPT-4o and Gemini 1.5 Pro, showcasing their current performance levels and identifying specific challenges they face. It effectively points out the significant gap between human capabilities and the current state-of-the-art in long-context video understanding, thus paving the way for targeted improvements in AI development.

**Weaknesses:**

The paper’s focus on evaluating models using video records, rather than interactive environments, introduces several limitations. This approach confines the assessment to passive information retrieval and task completion without testing an agent’s adaptive capabilities in real-time. Consequently, it fails to simulate dynamic, interactive challenges where agents need to respond to changing conditions and feedback, limiting the benchmark’s applicability for real-world scenarios. Additionally, video records do not fully capture the complexity of decision-making in live environments where agents must process incomplete or misleading information.

The paper does not specifically address potential biases related to the content of the videos used in the VideoWebArena benchmark, which raises concerns about the inclusivity and fairness of the evaluation. If the content records primarily reflect cultural, linguistic, or social contexts from a limited demographic, this could introduce biases that affect the model’s performance and generalization capabilities. Such biases may skew the results, favoring models trained on similar datasets while disadvantaging those that have been exposed to more diverse inputs. Additionally, there is the risk that the benchmark may not adequately represent ethnic and cultural variations in how information is presented or interpreted, limiting its applicability for global use cases. Addressing potential ethnic or cultural content biases is crucial for ensuring that the models evaluated can fairly and effectively serve users from diverse backgrounds.

**Questions:**

refer to above concerns

---

> ### Author Response · Authors · 2024-11-21
>
> We thank the reviewer for their time and insightful comments. We address each concern in turn below.
>
> **Re: dynamic, interactive challenges**
>
> * We agree that an interactive context is important in an agent’s ability to dynamically plan and accomplish tasks, which is exactly why our benchmark:
>   * **(1) evaluates the ability of agents in interactive environments** and
>   * **(2) evaluates the ability of agentic foundation models to process and understand information from a video modality in these dynamic settings.**
>
> * We emphasize that *our tasks all require deploying agents in a dynamic, interactive environment.* More information on our environment’s interactivity along with other details of the interactive environment and the tasks can be found in Section 3.
> * The live environments are all locally-hosted real websites that are populated with real data that can be modified by the agents’ actions.
> * These sites are responsive, changing in response to agent actions—just like how human users interact with websites.
> * The video information is provided in-context to the LLM agent at each time step as part of its input. Each of the agent’s actions causes a change in the webpage state; for example, clicking on a link will change the webpage while the agent is doing a task. We will make this clearer in our paper.
>
>
> **Re: potential biases**
>
> * We agree with the reviewer on the importance of inclusivity/fairness in benchmark datasets—while no benchmark can perfectly represent all global variations, we have made considerable efforts to ensure that ours is sufficiently broad to test robustness in our area of focus across realistic scenarios.
> * To clarify, our main focus is studying the ability of agents to make use of video information to accomplish tasks in dynamic settings. As the issues of multi-modal video-capable agent performance and in-context video understanding pose huge obstacles, we made this the main focus of our data/evaluation set-up.
> * Nonetheless, to better address these issues, we will make several changes:
>   * We will add Portuguese/Chinese/Korean variations to the video dataset---though not comprehensive, we intend this to be a start.
>     * Given the significant time/labor required to implement new websites and the main focus of our paper, adding non-English websites unfortunately is beyond the scope of this paper. We have, however, made it so that our benchmark and its videos can be easily accessed and spun-up by the open-source community to test/study potential biases.
>   * We will include an ethics statement along with a discussion on the limitations and disclaimers about the reviewer’s points to our paper, citing the lack of diversity in tasks and video data that may not span the global use cases mentioned above.
>   * We also recognize that benchmarks evolve with the community’s needs, identification of new flaws/biases, etc., and welcome feedback. Benchmarks are iterative tools and we encourage future work to build upon our efforts to further address issues of inclusivity and global applicability.
>     * We have intentionally left room for updates and extensions to the dataset to incorporate additional underrepresented content or address gaps identified by the community.
>     * Our work and its documentation make it very easy for users and the open-source community to create their own tasks customized with their own videos, websites, etc. that reflect specific variations of interest that may be underrepresented here. Additionally, existing websites can be used to create new tasks.
>     * Our repo will be available and monitored to also address potential issues and problems as they arise with implementation, testing, etc.
>
> Please let us know of any further questions/concerns!

---

> > ### Author Response · Authors · 2024-11-24
> > **Follow-Up to Rebuttal**
> >
> > Dear reviewer, thank you again for your valuable feedback. Since the rebuttal period is ending soon, we were wondering whether our changes have addressed your concerns. Please let us know and we will be happy to engage further.

---

> ### Comment · Area_Chair_2YTG · 2024-11-27
>
> Dear reviewer,
>
> Today is the last day for reviewers to ask questions to authors. Did the authors' rebuttal address your concern? Do you have any additional questions?

---

### Author Response · Authors · 2024-11-27
**Manuscript revisions**

## Manuscript Revisions
We thank the reviewers for all their insightful feedback. We have uploaded a revised version of our manuscript in response to reviewer feedback in an effort to further improve our work. Changes include (but are not limited to):

* Quality of life changes throughout the paper (e.g., fixing typos, adding content and revisions to improve flow/clarity)
* Phi-3.5V Model results (Section 5, Tables 6 and 8),
* Dataset Comparison table (Section 2, Table 1),
* Revised Table 5 with a more relevant Temporal Reasoning Example
* Revised and improved introduction (with more context and references)
* Revised and improved discussion section
* Revised ethics statement to address possible biases and mitigation efforts
* Added Appendix B on more details of failure modes, task documentation creation, etc.

---

### Meta-Review · Area_Chair_2YTG · 2024-12-23

**Metareview:**

This paper was reviewed by five experts in the field. Reviewer pi6N disappeared and didn't participate in the discussion despite multiple reminders from the AC and authors. The authors' rebuttal resolved most concerns, and the other four reviewers unanimously agreed to accept the paper.

The AC read the paper, the reviews, and the rebuttal. The AC believes the concerns from reviewer pi6N were well addressed by the rebuttal. Therefore, the decision is to recommend the paper for acceptance. The reviewers did raise some valuable suggestions in the discussion that should be incorporated in the final camera-ready version of the paper. The authors are encouraged to make the necessary changes to the best of their ability.

**Additional Comments On Reviewer Discussion:**

One reviewer (pi6N) disappeared during the discussion stage. The reviewers' concerns were well addressed by the rebuttal. The other four reviewers unanimously agreed to accept the paper.

Reviewer pi6N also flagged the need for Ethics Review but disappeared during the discussion stage and didn't provide any further detail regarding the rebuttal. The AC believes the concerns from reviewer pi6N were well addressed by the rebuttal and, therefore, decided to ignore this request.

---

### Decision · Program_Chairs · 2025-01-22

Accept (Poster)